# Preserving Ordinality in Diabetic Retinopathy Grading through a Distribution-Based Loss Function

Lena Stelter[1,2,4], Valentina Corbetta[2,3,4], Soufyan Lakbir[4,5,6], Regina Beets-Tan[2,3], Ricardo P.M. Cruz[7,8], Jaime S. Cardoso[7,8], and Wilson Silva[*2,4]

[1]Department of Physics and Astronomy, University of Heidelberg, Heidelberg, Germany
[2]Department of Radiology, The Netherlands Cancer Institute, Amsterdam, The Netherlands
[3]GROW School for Oncology and Developmental Biology, Maastricht University Medical Center, Maastricht, The Netherlands
[4]AI Technology for Life, Department of Information and Computing Sciences, Department of Biology, Utrecht University, Utrecht, The Netherlands
[5]Bioinformatics Section, Department of Computer Science, Vrije Universiteit Amsterdam, Amsterdam, The Netherlands
[6]Translational Gastrointestinal Oncology Group, Department of Pathology, Netherlands Cancer Institute, Amsterdam, The Netherlands
[7]Faculdade de Engenharia, Universidade do Porto, Porto, Portugal
[8]INESC TEC, Porto, Portugal
`w.j.dossantossilva@uu.nl`

## Abstract

Diabetic Retinopathy (DR) is a neurovascular complication of diabetes and the leading cause of blindness in adults in developed countries. Because DR progresses through ordered severity levels, its grading is naturally an ordinal classification problem. Yet, most deep learning methods treat it as a categorical task, disregarding the inherent class order and worsening performance under class imbalance.

In this work, we introduce a novel ordinal loss function that emphasizes the predictive tendencies of the whole model output rather than the class output probabilities individually. This design promotes unimodal predictions aligned with the underlying severity scale and is particularly robust to class imbalance. To place our method in context, we also evaluate a range of existing ordinal approaches on five publicly available DR datasets. with cross-entropy serving as a nominal baseline.

Extensive experiments demonstrate that our proposed loss function consistently preserves the ordinal structure of DR grades, even under severe imbalance, outperforming both nominal and alternative ordinal formulations.

The code is publicly available at https://github.com/Trustworthy-AI-UU-NKI/Ordinal-DR-Grading.

## 1 Introduction

Diabetic Retinopathy (DR) is a complication of both type 1 and type 2 diabetes, where prolonged elevated blood glucose levels damage retinal blood vessels. Depending on severity, DR can progress from mild vision impairment to complete blindness, and it is the leading cause of blindness in adults aged 20-74 in developed countries [2]. Severity is typically categorised into five stages: no DR, mild DR, moderate DR, severe DR, and proliferative DR. Severity grades are depicted in Figure 1. Diagnosis can be performed by an ophthalmologist through eye examinations or retinal photography, the latter enabling more cost-effective and scalable screening through expert annotation and, increasingly, Deep Learning (DL) methods [3].

A crucial aspect of DR grading is that these severity levels are not arbitrary categories, but follow a natural order. This makes DR grading an ordinal classification problem. However, most DL approaches treat it as a nominal multi-class task, ignoring the structural information in the class ordering [4]. While some ordinal methods have been explored, they remain far less common than nominal formulations. Standard classifiers assume independence between classes and optimize predictions toward one-hot targets, which may limit their ability to capture the progression of disease severity.

Formally, ordinal classification aims to learn a function $\vartheta : \mathcal{X} \to \mathcal{Y}$, mapping inputs $\mathbf{x} \in \mathcal{X} \subseteq \mathbb{R}^d$ to labels $\{c_0, ..., c_{K-1}\} \in \mathcal{Y}$, while respecting the well-defined order $c_0 \prec c_1 \prec ... \prec c_{K-1}$.

Recent research has emphasized the importance of unimodal output distributions for ordinal classification [5–7]. A unimodal posterior ensures that, given a predicted class $\hat{k}$, the neighboring classes $c_{\hat{k}-1}$ and $c_{\hat{k}+1}$ hold the next highest probabilities, with probabilities decaying monotonically as the class index moves away from $\hat{k}$. This behavior reflects the continuity of disease progression and has been shown to improve alignment with ordinal structure [7, 8].

---

*Corresponding Author.

Proceedings of the 7th Northern Lights Deep Learning Conference (NLDL), PMLR 307, 2026.



| No DR | Mild DR | Moderate DR | Severe DR | Proliferate DR |

**Figure 1.** Diabetic Retinopathy (DR) grading. Images from the Aptos [1] dataset for each severity level $c_k$: no DR, mild DR, moderate DR, severe DR and proliferate DR [2].

Another important aspect is the taxonomy of ordinal approaches. Methods can be broadly categorized into *non-unimodal*, *soft-unimodal*, and *hard-unimodal* [7]. While enforcing unimodality can strongly align predictions with ordinal assumptions, recent work has shown that promoting unimodality in a softer manner can achieve competitive or even superior performance, finding a balance between order preservation and predictive accuracy.

Finally, DR datasets, like many medical datasets, suffer from class imbalance. Mild and moderate cases are much more frequent than severe or proliferative stages, which complicates both model training and evaluation [9]. Despite its prevalence, the interaction between class imbalance and ordinality remains unexplored.

In this work, we address these challenges by proposing a new approach to ordinal DR grading. Specifically:

- We introduce a novel non-parametric ordinal loss function, Expectation Mean Squared Error (exp_MSE) that encourages unimodal model outputs with limited hyperparameter tuning. By computing the mean and variance of the categorical distribution, our formulation penalizes deviations from the true class in proportion to their distance, thereby preserving local ordinal structure. This design yields consistent performance across diverse DR datasets.

- We provide a comprehensive empirical analysis of ordinal prediction behavior using visualization-based diagnostics (t-SNE plots, confusion matrices, and distributional analyses), offering insight into how different ordinal losses shape model outputs.

- We analyze the impact of class imbalance through multiple imbalance measures, highlighting how dataset characteristics influence predictive structure and model reliability.

- We contribute an evaluation framework that connects quantitative performance with qualitative prediction behavior, encouraging more transparent assessment of ordinal methods.

## 2 Related Work

### 2.1 Deep Learning for Diabetic Retinopathy Grading

DL has been extensively applied to DR, addressing tasks ranging from binary classification (no DR vs. DR) to the full five-stage grading, and in some cases also lesion or vessel segmentation [10]. Most studies employ image pre-processing techniques such as data augmentation, resizing, cropping, denoising, contrast enhancement, or color normalization to mitigate limited dataset size and heterogeneous image quality [11].

Model architectures vary from custom Convolutional Neural Networks (CNNs) to established backbones such as ResNet, AlexNet, or VGG, often enhanced by transfer learning to improve generalization. Attention mechanisms and Transformer-based models have been increasingly integrated to highlight fine-grained retinal patterns, provide more interpretable predictions, and address generalization issues across heterogeneous datasets [12, 13]. Attention has also been explored as a mechanism to mitigate class imbalance. Furthermore, generative models such as autoencoders and GANs have been used to augment training sets and enhance image quality [10, 13].

Despite these advances, most DL approaches treat DR grading as a nominal classification problem, thereby ignoring the inherent ordering of severity stages. This motivates the exploration of ordinal methods, which explicitly incorporate class order into the learning process.

### 2.2 Ordinal Classification Methods

Ordinal classification has been widely studied in Machine Learning (ML) as a way to incorporate ordering information into class predictions. Traditional approaches include non-unimodal methods such as Ordinal Encoding (OE) [14], which reformulates the ordinal task as a set of binary classification problems. More recent research has focused on promoting unimodality in model outputs, either softly or strictly.

Soft-unimodal constraints encourage unimodal distributions through loss formulations. Examples include label smoothing approaches that soften one-

hot targets [5], Binomial Cross-Entropy (BCE) [15], or distance-based smoothing functions [7], which penalize deviations from the closest unimodal distribution.

Hard-unimodal constraints, in contrast, enforce unimodality by design, typically via architectural components. For instance, UnimodalNet (UN) [7] introduces a dedicated output layer that guarantees unimodal predictions regardless of the learned logits.

Taxonomies have emerged to organize these approaches into non-unimodal, soft-unimodal, and hard-unimodal methods [7], with different trade-offs between flexibility and strict adherence to ordinal assumptions. While promoting unimodality has shown strong empirical benefits, the balance between order preservation and predictive accuracy remains an open challenge, especially under class imbalance.

## 2.3 Ordinal Methods for Diabetic Retinopathy Grading

Several works have adapted ordinal methods specifically for DR grading. De la Torre et al. [16] introduced a quadratic weighted kappa loss, directly aligning optimization with ordinal agreement. Galdran et al. [17] proposed cost-sensitive regularization of Cross-Entropy (CE) to better capture the ordinal nature of DR stages. Araújo et al. [18] combined CE with an uncertainty-based regularizer: the network outputs both a class prediction and an uncertainty estimate, modeled as a Gaussian distribution with mean given by the prediction and variance given by the uncertainty.

Other approaches integrate hybrid architectures and loss functions. Ma et al. [19] combined CNN and Transformer components with a joint cross-entropy and weighted kappa loss to capture both local and global features while preserving ordinal structure. Tian et al. [20] employed a soft-labelling strategy with a metric loss to cluster features and focal loss to counter class imbalance.

More recent work has incorporated language and multimodal learning. CLIP-DR [21] leverages text prompts of DR grades and aligns image–text embeddings, thereby mitigating class imbalance and preserving ordinal relations. Building on OrdinalCLIP, it introduces rank-aware embeddings that explicitly encode ordinal constraints. The AOR-DR framework [22] extends this line with an autoregressive ordinal regression design, while Lawate et al. [23] proposed a multi-stage pipeline combining Efficient-Net for initial screening with a Transformer-based ordinal regression head, further supported by uncertainty estimation and optional lesion segmentation modules.

Together, these works highlight growing interest in integrating ordinal formulations into DR grading. While these approaches adapt ordinal methods specifically for DR grading, they are often highly tailored to particular datasets, incorporate additional modalities (e.g., text, segmentation), or optimize task-specific metrics such as quadratic kappa. In contrast, our goal is to benchmark general-purpose ordinal methods under controlled conditions across multiple datasets, providing a systematic comparison and introducing a novel loss designed to facilitate generalization beyond DR.

# 3 Materials and Methods

## 3.1 Mathematical Notation

Let $\mathcal{X} \subseteq \mathbb{R}^d$ denote the input space and $\mathcal{Y} = \{c_0, \ldots, c_{K-1}\}$ the set of ordered class labels with $c_0 \prec c_1 \prec \cdots \prec c_{K-1}$. Given an input $\mathbf{x} \in \mathcal{X}$, a model outputs posterior probabilities $\hat{\mathbf{y}} = [\hat{y}_0, \ldots, \hat{y}_{K-1}]$, where $\hat{y}_i = \hat{p}(c_i|\mathbf{x})$, obtained via the softmax function. From here on we refer to the softmax outputs as posterior probabilities, i.e. the model's estimated posterior distribution over classes given the input.

We denote the true class index by $k \in \{0, \ldots, K-1\}$ and the predicted class index by

$$\hat{k} = \operatorname*{argmax}_{i \in \{0, \ldots, K-1\}} \hat{y}_i. \tag{1}$$

## 3.2 Expectation Mean Squared Error (exp_MSE)

The intuition behind our approach is illustrated in Figure 2. Conventional nominal approaches, such as CE (Figure 2a), often produce dispersed posterior distributions that are not well aligned with the true class, as they optimize for one-hot targets without accounting for class order. In contrast, our proposed loss (Figure 2b) encourages predictions that are both centered on the correct class and concentrated around it, thereby reflecting the ordinal structure of the labels.

To formalize this intuition, we first assume a continuous approximation of the posterior distribution. Let $\mathbf{y}^*$ denote the predicted posterior, which we approximate by a Gaussian distribution:

$$\mathbf{y}^* \approx \mathcal{N}(i; \mu, \sigma^2), \quad i \in \mathbb{R}_{\geq 0}, \tag{2}$$

where $\mu$ and $\sigma^2$ denote the mean and variance of the distribution. Ideally, we want predictions that converge to the correct class index $k$ with vanishing variance, i.e.,

$$\mathbf{y}^* \to \lim_{\substack{\mu \to k \\ \sigma^2 \to 0}} \mathcal{N}(i; \mu, \sigma^2) = \delta(i - k), \tag{3}$$

where $\delta(i - k)$ is the Dirac delta centered at the true class.

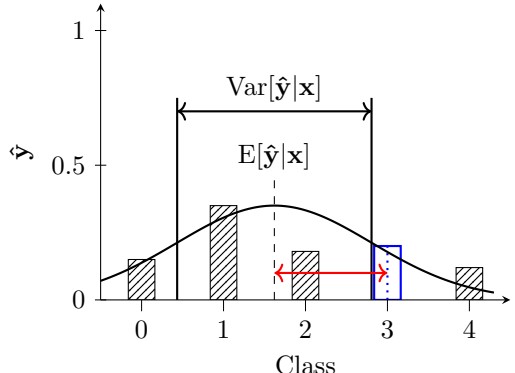

**(a)** When trained with a nominal loss such as cross-entropy (CE), the posterior may be dispersed and misaligned, with the expectation $E[\hat{\mathbf{y}}|\mathbf{x}]$ (dashed line) deviating from the true class and the variance $\mathrm{Var}[\hat{\mathbf{y}}|\mathbf{x}]$ remaining high.

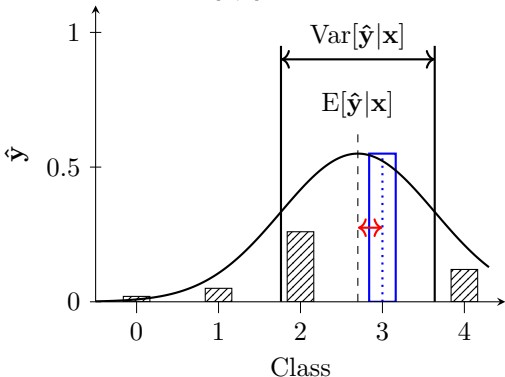

**(b)** When trained with the proposed exp_MSE loss, the distribution becomes centered on the true class and concentrated around it. Conceptually, this corresponds to approximating the posterior by a Gaussian $\mathcal{N}(\mu, \sigma^2)$ and driving it toward a Dirac delta $\delta(i-k)$, with $\mu \to k$ and $\sigma^2 \to 0$.

**Figure 2.** Intuition behind the proposed exp_MSE loss. The histogram shows the model's predicted posterior distribution $\hat{\mathbf{y}}$ over classes, with the true class at $k = 3$ indicated by the blue bin, approximated by a Gaussian $\mathcal{N}(\mu, \sigma^2)$ (black line) with $\mu = E[\hat{\mathbf{y}}|\mathbf{x}]$ and $\sigma^2 = \mathrm{Var}[\hat{\mathbf{y}}|\mathbf{x}]$.

Since ordinal classification is inherently discrete, we instead model the posterior as a categorical distribution $\hat{\mathbf{y}}$. In this setting, the mean and variance of the predicted distribution are given by

$$E[\hat{\mathbf{y}}|\mathbf{x}] = \sum_{i=0}^{K-1} \hat{y}_i \, i, \tag{4}$$

$$\mathrm{Var}[\hat{\mathbf{y}}|\mathbf{x}] = \sum_{i=0}^{K-1} \hat{y}_i \, (i - E[\hat{\mathbf{y}}|\mathbf{x}])^2. \tag{5}$$

We then define the exp_MSE loss as

$$\mathcal{L}_{\mathrm{exp\_MSE}}(\hat{\mathbf{y}}, k) = \left| E[\hat{\mathbf{y}}|\mathbf{x}] - k \right|^2 + \lambda \, \mathrm{Var}[\hat{\mathbf{y}}|\mathbf{x}], \tag{6}$$

where $\lambda$ controls the influence of the variance term. The first term ensures the mean prediction aligns with the true class, while the second term reduces uncertainty by penalizing wide distributions.

This formulation thus directly operationalizes the intuition: predictions should be accurate (mean close to $k$) and unimodal (low variance).

## 3.3  Baseline Methods

We compare our proposed exp_MSE loss against several established ordinal classification approaches, covering non-unimodal, soft-unimodal, and hard-unimodal strategies.

**Cross-Entropy (CE).** For multi-class nominal classification, the CE loss is the standard objective. Given a one-hot encoded label $\mathbf{y} \in \{0, 1\}^K$ and a model prediction $\hat{\mathbf{y}} \in [0, 1]^K$, CE is defined as

$$\mathcal{L}_{\mathrm{CE}}(\mathbf{y}, \hat{\mathbf{y}}) = -\sum_{i=0}^{K-1} y_i \log \hat{y}_i. \tag{7}$$

We treat CE as our nominal (non-unimodal) baseline.

**Ordinal Encoding (OE).** Frank et al. [14] proposed OE as a non-unimodal approach that reformulates the $K$-class problem into $K-1$ binary tasks. For a given true class $k$, the cumulative binary labels are defined as

$$y_i = \begin{cases} 1, & \text{if } i < k, \\ 0, & \text{otherwise,} \end{cases} \quad \text{for } i \in \{0, \dots, K-2\}. \tag{8}$$

Each binary classifier is trained with binary CE. At inference time, the class probabilities $q_i$ are reconstructed from the logits $g_i$ via the sigmoid $\sigma(x) = \frac{1}{1+\exp(-x)}$:

$$q_i = \begin{cases} 1 - \sigma(g_0), & \text{if } i = 0, \\ \sigma(g_{i-1}) - \sigma(g_i), & \text{if } 1 \le i \le K-2, \\ \sigma(g_{K-1}), & \text{if } i = K-1. \end{cases} \tag{9}$$

**Binomial Cross-Entropy (BCE).** Label smoothing [5] replaces the one-hot target with a softened distribution:

$$h'(i, k) = (1 - \eta)\delta_{i,k} + \eta f(i, k), \tag{10}$$

where $\eta \in [0, 1]$ controls smoothing and $f(i, k)$ is a distribution centered on the true class. In the BCE formulation [15], $f(i, k)$ is chosen as a binomial:

$$f(i, k) = \binom{K-1}{i} p_k^i (1 - p_k)^{K-1-i}, \quad i \in \{0, \dots, K-1\}, \tag{11}$$

with $p_k$ the success probability.

**$L_p$ Label Smoothing.** Vargas et al. [24] proposed an alternative smoothing function:

$$f(i, k) = \frac{\exp\left(-|i - k|^p\right)}{\sum_{j=0}^{K-1} \exp\left(-|j - k|^p\right)}, \quad 1 \le p \le 2. \tag{12}$$

Smaller values of $p$ produce broader distributions, while larger values concentrate mass closer to the true class.

**Wasserstein-Unimodal-Wasserstein (Wu-Wass).** Cardoso et al. [7] introduced a regularization scheme that encourages unimodality. The method projects the predicted distribution $\hat{\mathbf{y}}$ to the closest unimodal distribution $\hat{\mathbf{y}}^P$ with mode at $k$ using the Wasserstein distance $\mathcal{D}_1$:

$$\hat{\mathbf{y}}^P = \arg\min_{y \in \mathcal{S}} \mathcal{D}_1(\mathbf{y}, \hat{\mathbf{y}}), \qquad (13)$$

where $\mathcal{S}$ is the set of unimodal distributions with mode $k$. The loss is then defined as

$$\mathcal{L}_{\text{wu\_wass}}(\mathbf{y}, \hat{\mathbf{y}}) = \mathcal{L}_{\text{CE}}(\mathbf{y}, \hat{\mathbf{y}}) + \lambda \, \mathcal{D}_2(\hat{\mathbf{y}}^P, \hat{\mathbf{y}}), \quad (14)$$

where $\mathcal{D}_2$ is typically instantiated as Wasserstein distance or KL divergence.

**Mean-Variance Loss (MVL)** The Mean-Variance Loss (MVL) introduced by Pan et al. [25] extends CE by incorporating additional mean and variance terms:

$$\mathcal{L}_m = (\mathrm{E}[\hat{\mathbf{y}}|\mathbf{x}] - k)^2 \qquad (15)$$
$$\mathcal{L}_v = \mathrm{Var}[\hat{\mathbf{y}}|\mathbf{x}] \qquad (16)$$

as defined in Equations 4 and 5. The final objective combines these components with two hyperparameters $\lambda_1$ and $\lambda_2$:

$$\mathcal{L}_{\text{MVL}} = \mathcal{L}_{\text{CE}} + \tfrac{\lambda_1}{2}\mathcal{L}_m + \lambda_2\mathcal{L}_v. \qquad (17)$$

Unlike our method, MVL uses mean and variance solely as regularizers.

**UnimodalNet (UN).** Finally, a hard-unimodal approach is UN [7], which enforces unimodality by construction. Given model outputs $\mathbf{v} \in \mathbb{R}^K$, non-negative activations $z_i = f(v_i)$ are computed. Two cumulative sums are then formed:

$$z_0^{\ell r} = z_0, \qquad z_i^{\ell r} = z_{i-1}^{\ell r} + z_i, \;\; i = 1, \ldots, K-1, \qquad (18)$$

$$z_{K-1}^{r\ell} = z_{K-1}, \quad z_i^{r\ell} = z_{i+1}^{r\ell} + z_i, \;\; i = K-2, \ldots, 0. \qquad (19)$$

The elementwise minimum $z_i^u = \min(z_i^{\ell r}, z_i^{r\ell})$ ensures unimodality, and the final prediction is obtained via softmax:

$$\hat{\mathbf{y}} = \text{softmax}(\mathbf{z}^u). \qquad (20)$$

## 3.4 Datasets

We evaluate our method on five publicly available DR datasets: Aptos [1], IDRiD [26], DeepDR [27],

**Table 1.** Overview of the Diabetic Retinopathy (DR) datasets, including training, validation and test sample counts and total sample counts.

| Dataset | Train | Validation | Test | Total |
|---|---|---|---|---|
| DeepDR [27] | 1278 | 322 | 400 | 2000 |
| IDRiD [26] | 328 | 85 | 103 | 516 |
| Aptos [1] | 2334 | 587 | 741 | 3662 |
| RLDR [28] | 1023 | 258 | 312 | 1593 |
| DDR [29] | 7008 | 1754 | 3759 | 12521 |

**Table 2.** Imbalance statistics for the five DR datasets, reported using the Imbalance Ratio (IR), the Min–Max Imbalance Ratio (IR$_{\text{minmax}}$), and the normalized Shannon entropy ($H$). Datasets are ordered from least to most imbalanced according to IR. Arrows indicate the direction of increasing imbalance.

| Dataset | IR ↑ | IR$_{\text{minmax}}$ ↓ | Shannon ↓ |
|---|---|---|---|
| DeepDR [27] | 1.691 | 0.129 | 0.873 |
| IDRiD [26] | 1.789 | 0.147 | 0.894 |
| Aptos [1] | 2.169 | 0.101 | 0.797 |
| RLDR [28] | 2.506 | 0.073 | 0.740 |
| DDR [29] | 4.325 | 0.038 | 0.702 |

RLDR [28], and DDR [29]. For consistency, we adopt the pre-processing and splits provided by GDRBench [4], including black border removal, background masking, and resizing to 512×512 pixels. Table 1 summarizes the dataset statistics.

**Data Augmentation.** To improve generalization, we apply several augmentation techniques during training. Specifically, each image is randomly rotated by 90° with probability 0.5, and random horizontal and vertical flips are applied with the same probability. Color jittering is performed with brightness, contrast, saturation, and hue adjustments of 0.3. We also apply random resized cropping, where the image is randomly scaled before being center-cropped. All images are resized to $224 \times 224$ pixels and normalized to ImageNet's channel-wise mean and standard deviation.

## 3.5 Quantifying Class Imbalance

To characterize the imbalance in the DR datasets, we report three complementary metrics: the Imbalance Ratio (IR), the Min–Max Imbalance Ratio (IR$_{\text{minmax}}$), and the normalized Shannon entropy ($H$).

**Imbalance Ratio (IR).** The IR [30] measures the average pairwise imbalance across all classes $K$

and is defined as

$$\text{IR} = \frac{1}{K} \sum_{i=0}^{K-1} \frac{N - N_i}{(K-1)N_i}, \qquad (21)$$

where $N$ is the total number of samples and $N_i$ the number of samples in class $i$. A balanced dataset yields $\text{IR} = 1$, while higher values indicate increasing imbalance. Following Perez et al. [30], we consider datasets with $\text{IR} > 1.5$ as highly imbalanced.

**Min–Max Imbalance Ratio ($\text{IR}_{\text{minmax}}$).** To capture the imbalance between the majority and minority class, we compute the min–max ratio [31]:

$$\text{IR}_{\text{minmax}} = \frac{\min_i N_i}{\max_i N_i}. \qquad (22)$$

Here, values close to 1 correspond to balanced datasets, while lower ratios indicate greater imbalance.

**Normalized Shannon Entropy ($H$).** The normalized Shannon entropy measures the uncertainty of the class distribution:

$$H = -\frac{1}{\log(K)} \sum_{i=0}^{K-1} \frac{N_i}{N} \log\left(\frac{N_i}{N}\right). \qquad (23)$$

This metric ranges between 0 and 1, where $H = 1$ indicates a perfectly balanced dataset (uniform distribution) and $H = 0$ corresponds to complete imbalance (all samples concentrated in a single class).

The imbalance statistics for the five DR datasets are reported in Table 2.

## 3.6 Evaluation Metrics

In ordinal classification, evaluation should capture both nominal classification quality and the ordinal structure of the labels. To this end, we report the following metrics:

- **Balanced Accuracy (BA)**: evaluates classification quality across all classes by averaging recall per class, ensuring robustness against imbalance.

- **Average Mean Absolute Error (AMAE)** [32]: measures the average distance between true and predicted classes across all labels, providing a distance-based error robust to class imbalance.

- **Kendall's $\tau_b$** [34]: quantifies the rank correlation between true and predicted labels, assessing whether predictions preserve ordinal ordering.

- **Uniform Ordinal Classification Index ($A_{\text{UOC}}$)** [33, 35]: a composite ordinal metric that integrates both ranking agreement and distance-based penalties, and accounts for class imbalance.

Mathematical formulations of these metrics are provided in Appendix A.

## 3.7 Model Architecture and Training Settings

We adopt a ResNet-50 [36] initialized with ImageNet [37] pre-trained weights. The training set provided by GDRBench is further split into training and validation subsets using a ratio of 80:20.

We train for 100 epochs using five train/validation splits (seeds 0-4). Optimization is performed using the Adam [38] optimizer with a learning rate of 0.001 and a batch size of 64.

Hyperparameters for exp_MSE, Wasserstein-Unimodal-Wasserstein (Wu-Wass) and $L_p$ losses are tuned using Optuna [39] on the validation sets. Details of the search ranges and the final hyperparameters are reported in B.

## 4 Results

### 4.1 Quantitative Results

Table 3 summarizes the averaged performance across the five DR datasets. Dataset-specific results are provided in the Appendix (Tables C.1–C.2). For each metric, the best results are shown in bold and the second-best results are underlined.

On average, our proposed method achieves the best performance across the three ordinal metrics AMAE, $A_{\text{UOC}}$, and Kendall's $\tau_b$, while ranking second in BA behind the soft-unimodal Wu-Wass method. The $L_p$ approach consistently provides the second-best ordinal results. Interestingly, not all ordinal-aware methods outperform the nominal CE baseline: BCE, UN, and MVL yield weaker performance, with UN performing worst overall. While MVL performs well across datasets, albeit still below our proposed method, its weaker results may reflect challenges in effectively tuning its two hyperparameters.

The $A_{\text{UOC}}$ score, which integrates ranking quality, error distance, and class imbalance, is the most comprehensive ordinal measure. According to $A_{\text{UOC}}$, exp_MSE achieves the best results on Aptos (Table C.1), RLDR (Table C.3), and DeepDR (tied with $L_p$, see Table C.4), the second-best score on DDR (after $L_p$, see Table C.2), and the third-best score on IDRiD (behind Wu-Wass and CE).

**Table 3.** Averaged results for all five DR datasets. Performance in terms of the Average Mean Absolute Error (AMAE) [32], Uniform Ordinal Classification Index ($A_{\mathrm{UOC}}$) [33], Kendall's $\tau_b$ [34] and the Balanced Accuracy (BA), comparing: Cross-Entropy (CE), Ordinal Encoding (OE) [14], Binomial Cross-Entropy (BCE) [5], UnimodalNet (UN) [7], $L_p$ [6], Wasserstein-Unimodal-Wasserstein (Wu-Wass) [7], and Mean-Variance Loss (MVL) [25] to our proposed method exp_MSE. Arrows indicate the direction of better performance for each metric. The best results are reported in **bold**, the second best results are underlined.

| Experiment | AMAE ↓ | $A_{\mathrm{UOC}}$ ↓ | Kendall's $\tau_b$ ↑ | BA ↑ |
|---|---|---|---|---|
| CE | $0.668 \pm 0.059$ | $0.631 \pm 0.030$ | $0.674 \pm 0.025$ | $0.518 \pm 0.035$ |
| OE | $0.653 \pm 0.051$ | $0.626 \pm 0.029$ | $0.675 \pm 0.031$ | $0.520 \pm 0.033$ |
| BCE | $0.659 \pm 0.030$ | $0.634 \pm 0.017$ | $0.680 \pm 0.025$ | $0.509 \pm 0.026$ |
| UN | $0.766 \pm 0.041$ | $0.675 \pm 0.019$ | $0.659 \pm 0.033$ | $0.465 \pm 0.025$ |
| $L_p$ | $0.648 \pm 0.035$ | $0.622 \pm 0.017$ | $0.683 \pm 0.016$ | $0.522 \pm 0.021$ |
| Wu-Wass | $0.652 \pm 0.034$ | $0.628 \pm 0.014$ | $0.676 \pm 0.022$ | $0.520 \pm 0.018$ |
| MVL | $0.667 \pm 0.034$ | $0.630 \pm 0.018$ | $0.670 \pm 0.023$ | $0.518 \pm 0.021$ |
| exp_MSE | $\mathbf{0.615 \pm 0.046}$ | $\mathbf{0.609 \pm 0.027}$ | $\mathbf{0.695 \pm 0.023}$ | $\mathbf{0.535 \pm 0.031}$ |

## 4.2 Impact of Class Imbalance

All datasets exhibit substantial class imbalance, quantified by IR, $\mathrm{IR}_{\mathrm{minmax}}$, and Shannon entropy $H$ (Table 2). Our method achieves the best results on three of the four most imbalanced datasets (by IR), and second-best on DDR, the most imbalanced dataset. Performance relative to $\mathrm{IR}_{\mathrm{minmax}}$ indicates robustness to strong minority/majority discrepancies, while results on Shannon entropy $H$ suggest that datasets with more uniform distributions (e.g. IDRiD) remain more challenging.

## 4.3 Confusion Matrices and Class Distributions

Figures C.1–C.9 in the Appendix visualize the confusion matrices and class distributions. Well-performing methods are expected to exhibit concentrated diagonals with errors distributed to adjacent classes. On Aptos and DeepDR, both $L_p$ and exp_MSE display the clearest diagonal structures, reflecting strong ordinal consistency. On DDR, the most imbalanced dataset, all methods show dispersion, but exp_MSE maintains a more stable diagonal. For IDRiD, no method preserves the ordinal structure strongly, reflecting its relatively balanced distribution and higher entropy.

## 4.4 Feature Analysis

To further analyze the learned representations, we applied t-SNE [40] to the feature embeddings (Figures C.11–C.15 in the Appendix). Datasets such as Aptos (classes 1-4), DDR, and DeepDR exhibit clear ordinal gradients across classes, while RLDR shows weak ordering and IDRiD shows almost none.

exp_MSE generally preserves ordinal gradients as well as or better than other baselines, consistent with its quantitative performance.

## 4.5 Posterior Distributions

To better understand the predictive behavior of the different methods, we visualize the normalized posterior distributions for each class across all samples of the DeepDR dataset (Figure 3). This provides an aggregated view of how strongly and consistently each method concentrates probability mass around the true class. Similar plots for the remaining datasets are reported in Appendix C.4.

As expected, the two soft-labelling approaches (BCE and $L_p$) produce unimodal posteriors that reflect the smoothed target distributions, often leading to broader probability spreads. In contrast, exp_MSE exhibits sharper and more concentrated posteriors, with smaller variance across classes, consistent with the variance-penalization in its formulation. This effect is especially visible in minority classes, where exp_MSE predictions remain more peaked compared to the other methods.

## 5 Conclusion

In this work, we evaluated ordinal methods for DR grading, focusing on approaches that encourage unimodal model outputs. We proposed a novel soft-unimodal loss, exp_MSE, which penalizes both deviations of the posterior mean from the true class and high variance in the predicted distribution.

Across five publicly available DR datasets, all of which exhibit substantial class imbalance, exp_MSE consistently outperformed baseline methods on the three ordinal metrics AMAE, $A_{\mathrm{UOC}}$, and Kendall's $\tau_b$. It also ranked second in BA, despite this metric being outside the primary scope of ordinal evaluation. While some baselines occasionally surpassed exp_MSE on individual datasets or metrics, none matched its overall consistency across all experiments. Posterior distribution analyses further confirmed that exp_MSE yields sharper, lower-variance

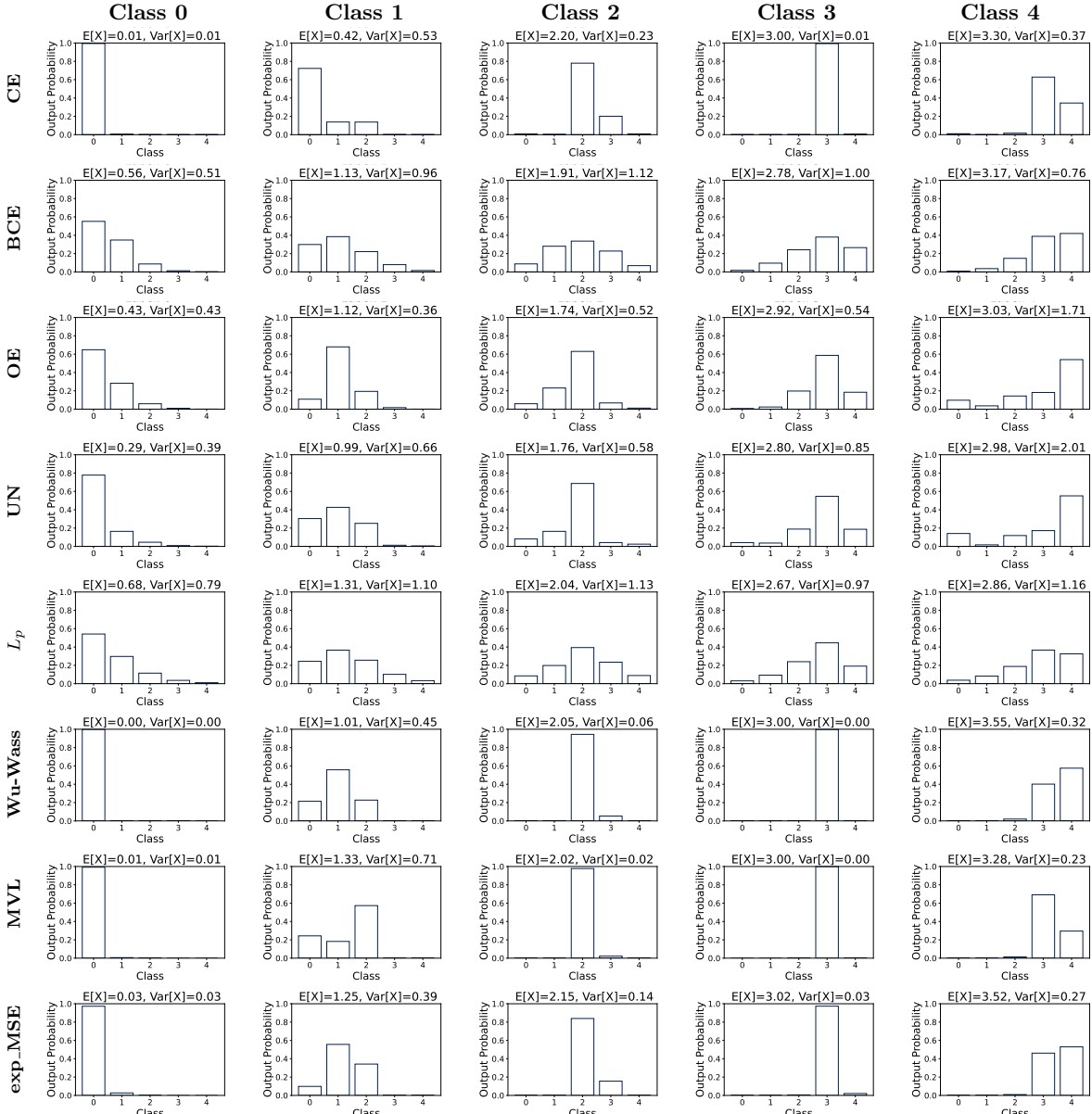

**Figure 3.** Normalized posterior distributions on the DeepDR dataset for the train/validation split seed 0, shown per true class (columns) comparing the methods (rows): Cross-Entropy (CE), Ordinal Encoding (OE) [14], Binomial Cross-Entropy (BCE) [5], UnimodalNet (UN) [7], $L_p$ [6], Wasserstein-Unimodal-Wasserstein (Wu-Wass) [7], and Mean-Variance Loss (MVL) [25] to our proposed method exp_MSE.

predictions compared to soft-labelling approaches such as BCE and $L_p$. Feature-space analysis via t-SNE showed that exp_MSE preserves ordinal gradients in most datasets, performing comparably or better than competing methods even in challenging scenarios of high imbalance.

**Limitations and Future Work.** While effective, the current formulation of exp_MSE affects classes differently depending on their position in the ordinal scale. Edge classes tend to be penalized with stronger variance constraints, whereas inner classes encourage more symmetric distributions. Future research could address this limitation through adaptive normalization strategies or by exploring alter-

native class topologies (e.g., circular arrangements) to ensure more uniform treatment of boundary and intermediate classes.

**Summary.** Overall, exp_MSE provides a robust and effective solution for ordinal DR grading, particularly under class imbalance. Its consistent performance across multiple datasets and evaluation metrics highlights its potential for broader applications in medical image analysis and other domains where ordinal structure plays a central role.

# Acknowledgments

Research at the Netherlands Cancer Institute is supported by grants from the Dutch Cancer Society and the Dutch Ministry of Health, Welfare and Sport. The authors would like to acknowledge the Research High Performance Computing (RHPC) facility of the Netherlands Cancer Institute (NKI). This publication is part of the project "Ordinality-informed Federated Learning for Robust and Explainable Radiology AI" with file number NGF.1609.241.009 of the research programme AiNED XS Europa which is (partly) financed by the Dutch Research Council (NWO).

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

# A  Ordinal Metrics

In ordinal classification, the evaluation of ordinality in the data involves both ranking agreement and distance-based error (distance between true and predicted class).

**Average Mean Absolute Error (AMAE).** The AMAE [32] determines the error of each class separately for each class and then gives the average error, quantifying the uniformity of the prediction: [33]

$$\text{MAE}_i = \frac{1}{N} \sum_{j=0}^{N-1} |k^j - \hat{k}^j| \tag{24}$$

$$\text{AMAE} = \frac{1}{K} \sum_{i=0}^{K-1} \text{MAE}_i \tag{25}$$

This metric is also suitable for imbalanced data. However, it overlooks the importance of ranking, which is central to ordinal tasks [33]. It is worth noting that for a perfectly balanced dataset the AMAE agrees with the Mean Absolute Error (MAE) [32], thus, making the MAE more robust to class imbalance [33].

**Kendall's $\tau_b$.** The Kendall's $\tau_b$ [34] measures the association between predicted and true rankings, reflecting the relative order of class predictions, but ignoring the magnitude of errors. In particular, it is independent of the numerical encoding of each class [33]. The score lies in $[-1, 1]$, where $\tau_b = 1$ indicates perfect rank agreement, $\tau_b = -1$ complete rank inversion and $\tau_b = 0$ no rank correlation [41].

$$\tau_b = \frac{\sum q_{ij} p_{ij}}{\sqrt{\sum q_{ij}^2 \sum p_{ij}^2}}, \tag{26}$$

Here, $q_{ij} = 1$ if $q_i > q_j$, $q_{ij} = 0$ if $q_i = q_j$ and $q_{ij} = -1$ if $q_i < q_j$, $p_{ij}$ is computed analogously. These value form the ranking matrices $Q$ (ground truth) and $P$ (prediction), which are asymmetric ($q_{ij} = -q_{ji}$ and $p_{ij} = -p_{ji}$) [33–35].

**Uniform Ordinal Classification Index ($A_{\text{UOC}}$).** The $A_{\text{UOC}}$ [33] is based on the Uniform Ordinal Classification Index (UOC$_{\beta}^{\gamma}$), and evaluates ordinal classification performance by capturing both ranking quality and distance-based error. This is done by traversing through the confusion matrix $M$ from entry $M_{0,0}$ to entry $M_{K-1,K-1}$, characterizing each possible path by a penalty or a benefit. Benefits are given to paths with many correct predictions, while penalties are assigned for deviations from the diagonal, depending on the distance between true and predicted class.

In addition to assessing magnitude of error and ranking, the metric also generalizes to class imbalance and unobserved classes in a dataset. The metric brings robustness to class imbalance by adding a prior $p(k)$:

$$p(k) = \frac{1}{K} \tag{27}$$

to the metric, similarly as the AMAE brings robustness for class imbalance for MAE. Unobserved classes are accounted for by an indicator function

$$\mathbb{1}_{\Omega}(k) = \begin{cases} 1, & \text{if } k \in \Omega \\ 0, & \text{if } k \notin \Omega \end{cases} \tag{28}$$

in a set of observed classes $\Omega$. To account for both class imbalance and unobserved classes, both assumptions can be combined to a total prior $p(k)$ by

$$p(k) = \frac{1}{K'} \mathbb{1}_{\Omega}(k), \tag{29}$$

where $K' \leq K$ defines the number of observed classes of set $\Omega$, i.e. $K' = |\Omega|$. Under these settings, the UOC$_{\beta}^{\gamma}$ is defined as

$$\text{UOC}_{\beta}^{\gamma} = \min \left\{ 1 - \frac{\sum_{(k,\hat{k}) \in \text{path}} p(\hat{k}|k) \mathbf{1}_{\mathcal{O}(k)}}{K' + \frac{K'}{K'^{\gamma}} \left( \sum_{(k,\hat{k})} p(\hat{k}|k) \mathbf{1}_{\mathcal{O}(k)} |k - \hat{k}|^{\gamma} \right)^{1/\gamma}} + \frac{\beta}{K'} \sum_{(k,\hat{k}) \in \text{path}} p(\hat{k}|k) \mathbf{1}_{\mathcal{O}(k)} |k - \hat{k}|^{\gamma} \right\}. \tag{30}$$

**Table B.1.** For each dataset, the best hyperparameters were obtained by Optuna [39]: $\lambda$ for exp_MSE (Equation 6), $\lambda$ for Wu-Wass (Equation 14) and $p$ for $L_p$ (Equation 12) and $\lambda_1, \lambda_2$ for MVL.

| Dataset | exp_MSE | $L_p$ | Wu-Wass | MVL |
|---|---|---|---|---|
| IDRiD [26] | $\lambda = 0.0867573447798667$ | $p = 1.82602171336898$ | $\lambda = 0.9118427146809470$ | $\lambda_1 = 0.800045755932323500$ |
| | | | | $\lambda_2 = 0.407117010948758200$ |
| RLDR [28] | $\lambda = 0.0465538229994773$ | $p = 1.30201892088105$ | $\lambda = 3.1668410016395200$ | $\lambda_1 = 0.538463844037895000$ |
| | | | | $\lambda_2 = 0.661145716810206400$ |
| DeepDR [27] | $\lambda = 0.0400201073104795$ | $p = 1.08456996367382$ | $\lambda = 0.0527391141324065$ | $\lambda_1 = 0.363572972291989700$ |
| | | | | $\lambda_2 = 0.318344694660245300$ |
| Aptos [1] | $\lambda = 0.0189014846330842$ | $p = 1.29425445876255$ | $\lambda = 1.9747675885985000$ | $\lambda_1 = 0.011839109647973772$ |
| | | | | $\lambda_2 = 0.903223322149407700$ |
| DDR [29] | $\lambda = 0.1049552111518660$ | $p = 1.35477076647998$ | $\lambda = 0.8850007162074970$ | $\lambda_1 = 0.965743024434928500$ |
| | | | | $\lambda_2 = 0.633033834277408200$ |

For this formulation, $\beta$ is a tunable parameter controlling the balance between ranking accuracy and distance-based error. For $\beta \geq 1$, the index prioritizes correctness over ordinal structure. For ideal evaluation $\beta$ should be in the interval of [0,1], where $\beta \to 0$ emphasizes ranking and $\beta \to 1$ prioritizes precise predictions [33, 42]. Parameter $\gamma$ controls the penalty of distance error. The $A_{\mathrm{UOC}}$ uses $\gamma = 1$, thus a linearly increasing error distance.

Finally, the $A_{\mathrm{UOC}}$ can be obtained by

$$A_{\mathrm{UOC}} = \int_0^1 \mathrm{UOC}_1^\beta \, d\beta, \tag{31}$$

therefore, removing the sensitivity of the $\beta$ parameter [33].

# B    Hyperparameters Fine-tuning

To tune the hyperparameters of the Wu-Wass, $L_p$, MVL and exp_MSE loss functions, we use Optuna [39], a library specifically designed for hyperparameter optimization via Bayesian processes. Each tuning was performed across all five train/validation split seeds for 25 epochs. The $A_{\mathrm{UOC}}$ scores for each final epoch of a train/validation split seed was then averaged for all seeds. The hyperparameter was chosen for the lowest average $A_{\mathrm{UOC}}$ score from the final epoch and then averaged across all five folds. Final hyperparameter selections are shown in Table B.1.

The search intervals we used:

- **exp_MSE**: $\lambda \in [0.001, 1.0]$, interval was set logarithmically

- **Wu-Wass**: $\lambda \in [0.01, 100.0]$, interval was set logarithmically

- **$L_p$**: $p \in [1, 2]$

- **MVL**: $\lambda_1, \lambda_2 \in [0, 1]$

The hyperparameter $\eta$ in Equation 10 for BCE and $L_p$, is set to 1 as given as default in the `dlordinal` library [15].

# C  Additional Results

## C.1  Quantitative Results

**Table C.1.** Results on the Aptos dataset. Performance in terms of the Average Mean Absolute Error (AMAE) [32], Uniform Ordinal Classification Index ($A_{\text{UOC}}$) [33], Kendall's $\tau_b$ [34] and the Balanced Accuracy (BA), comparing: Cross-Entropy (CE), Ordinal Encoding (OE) [14], Binomial Cross-Entropy (BCE) [5], UnimodalNet (UN) [7], $L_p$ [6], Wasserstein-Unimodal-Wasserstein (Wu-Wass) [7], and Mean-Variance Loss (MVL) [25] to our proposed method exp_MSE. Arrows indicate the direction of better performance for each metric. The best results are reported in **bold**, the second best results are underlined.

| Experiment | AMAE ↓ | $A_{\text{UOC}}$ ↓ | Kendall's $\tau_b$ ↑ | BA ↑ |
|---|---|---|---|---|
| CE | $0.505 \pm 0.029$ | $0.538 \pm 0.019$ | $0.849 \pm 0.012$ | $0.632 \pm 0.015$ |
| OE | $0.489 \pm 0.027$ | $0.527 \pm 0.018$ | $\mathbf{0.856 \pm 0.007}$ | $0.635 \pm 0.021$ |
| BCE | $0.510 \pm 0.028$ | $0.551 \pm 0.021$ | $0.853 \pm 0.006$ | $0.605 \pm 0.024$ |
| UN | $0.662 \pm 0.033$ | $0.624 \pm 0.015$ | $0.816 \pm 0.007$ | $0.541 \pm 0.023$ |
| $L_p$ | $0.485 \pm 0.035$ | $0.527 \pm 0.022$ | $\underline{0.854 \pm 0.004}$ | $\underline{0.639 \pm 0.026}$ |
| Wu-Wass | $0.491 \pm 0.034$ | $0.527 \pm 0.016$ | $\underline{0.854 \pm 0.013}$ | $0.638 \pm 0.012$ |
| MVL | $\underline{0.484 \pm 0.009}$ | $\underline{0.523 \pm 0.007}$ | $\mathbf{0.856 \pm 0.007}$ | $\mathbf{0.642 \pm 0.009}$ |
| exp_MSE | $\mathbf{0.471 \pm 0.033}$ | $\mathbf{0.519 \pm 0.021}$ | $0.853 \pm 0.012$ | $0.638 \pm 0.013$ |

**Table C.2.** Results on the DDR dataset. Performance in terms of the Average Mean Absolute Error (AMAE) [32], Uniform Ordinal Classification Index ($A_{\text{UOC}}$) [33], Kendall's $\tau_b$ [34] and the balanced accuracy, comparing: Cross-Entropy (CE), Ordinal Encoding (OE) [14], Binomial Cross-Entropy (BCE) [5], UnimodalNet (UN) [7], $L_p$ [6], Wasserstein-Unimodal-Wasserstein (Wu-Wass) [7], and Mean-Variance Loss (MVL) [25] to our proposed method exp_MSE. Arrows indicate the direction of better performance for each metric. The best results are reported in **bold**, the second best results are underlined.

| Experiment | AMAE ↓ | $A_{\text{UOC}}$ ↓ | Kendall's $\tau_b$ ↑ | BA ↑ |
|---|---|---|---|---|
| CE | $0.549 \pm 0.040$ | $0.562 \pm 0.024$ | $0.752 \pm 0.021$ | $\mathbf{0.585 \pm 0.019}$ |
| OE | $0.560 \pm 0.036$ | $0.574 \pm 0.021$ | $0.761 \pm 0.020$ | $0.566 \pm 0.021$ |
| BCE | $0.545 \pm 0.013$ | $0.571 \pm 0.008$ | $0.768 \pm 0.004$ | $0.563 \pm 0.012$ |
| UN | $0.641 \pm 0.017$ | $0.624 \pm 0.010$ | $0.747 \pm 0.013$ | $0.500 \pm 0.009$ |
| $L_p$ | $\mathbf{0.522 \pm 0.010}$ | $\mathbf{0.552 \pm 0.007}$ | $\mathbf{0.774 \pm 0.007}$ | $\mathbf{0.585 \pm 0.008}$ |
| Wu-Wass | $0.571 \pm 0.030$ | $0.577 \pm 0.018$ | $0.755 \pm 0.011$ | $0.570 \pm 0.021$ |
| MVL | $0.566 \pm 0.033$ | $0.575 \pm 0.018$ | $0.754 \pm 0.006$ | $0.572 \pm 0.013$ |
| exp_MSE | $\underline{0.526 \pm 0.066}$ | $\underline{0.561 \pm 0.047}$ | $\underline{0.769 \pm 0.006}$ | $\underline{0.579 \pm 0.060}$ |

**Table C.3.** Results on the RLDR dataset. Performance in terms of the Average Mean Absolute Error (AMAE) [32], Uniform Ordinal Classification Index ($A_{\text{UOC}}$) [33], Kendall's $\tau_b$ [34] and the balanced accuracy, comparing: Cross-Entropy (CE), Ordinal Encoding (OE) [14], Binomial Cross-Entropy (BCE) [5], UnimodalNet (UN) [7], $L_p$ [6], Wasserstein-Unimodal-Wasserstein (Wu-Wass) [7], and Mean-Variance Loss (MVL) [25] to our proposed method exp_MSE. Arrows indicate the direction of better performance for each metric. The best results are reported in **bold**, the second best results are underlined.

| Experiment | AMAE ↓ | $A_{\text{UOC}}$ ↓ | Kendall's $\tau_b$ ↑ | BA ↑ |
|---|---|---|---|---|
| CE | $0.899 \pm 0.064$ | $0.762 \pm 0.030$ | $0.413 \pm 0.014$ | $0.360 \pm 0.058$ |
| OE | $0.878 \pm 0.071$ | $0.756 \pm 0.026$ | $0.427 \pm 0.075$ | $0.367 \pm 0.036$ |
| BCE | $\underline{0.825 \pm 0.022}$ | $\underline{0.743 \pm 0.015}$ | $\underline{0.467 \pm 0.038}$ | $\underline{0.383 \pm 0.032}$ |
| UN | $0.956 \pm 0.074$ | $0.774 \pm 0.028$ | $0.400 \pm 0.077$ | $0.333 \pm 0.038$ |
| $L_p$ | $0.862 \pm 0.049$ | $0.748 \pm 0.020$ | $0.435 \pm 0.042$ | $0.384 \pm 0.033$ |
| Wu-Wass | $0.849 \pm 0.041$ | $0.747 \pm 0.012$ | $0.420 \pm 0.039$ | $0.379 \pm 0.010$ |
| MVL | $0.884 \pm 0.038$ | $0.758 \pm 0.011$ | $0.441 \pm 0.027$ | $0.364 \pm 0.013$ |
| exp_MSE | $\mathbf{0.799 \pm 0.077}$ | $\mathbf{0.720 \pm 0.032}$ | $\mathbf{0.485 \pm 0.064}$ | $\mathbf{0.411 \pm 0.042}$ |

**Table C.4.** Results on the DeepDR dataset. Performance in terms of the Average Mean Absolute Error (AMAE) [32], Uniform Ordinal Classification Index ($A_{\mathrm{UOC}}$) [33], Kendall's $\tau_b$ [34] and the balanced accuracy, comparing: Cross-Entropy (CE), Ordinal Encoding (OE) [14], Binomial Cross-Entropy (BCE) [5], UnimodalNet (UN) [7], $L_p$ [6], Wasserstein-Unimodal-Wasserstein (Wu-Wass) [7], and Mean-Variance Loss (MVL) [25] to our proposed method exp_MSE. Arrows indicate the direction of better performance for each metric. The best results are reported in **bold**, the second best results are underlined.

| Experiment | AMAE ↓ | $A_{\mathrm{UOC}}$ ↓ | Kendall's $\tau_b$ ↑ | BA ↑ |
|---|---|---|---|---|
| CE | $0.594 \pm 0.120$ | $0.578 \pm 0.063$ | $0.752 \pm 0.039$ | $0.590 \pm 0.061$ |
| OE | $0.524 \pm 0.078$ | $0.548 \pm 0.052$ | $0.754 \pm 0.023$ | $0.619 \pm 0.053$ |
| BCE | $0.541 \pm 0.027$ | $0.557 \pm 0.017$ | $0.768 \pm 0.016$ | $0.611 \pm 0.021$ |
| UN | $0.651 \pm 0.036$ | $0.611 \pm 0.021$ | $0.736 \pm 0.016$ | $0.562 \pm 0.027$ |
| $L_p$ | $\mathbf{0.493 \pm 0.040}$ | $\mathbf{0.528 \pm 0.028}$ | $0.783 \pm 0.012$ | $\mathbf{0.633 \pm 0.023}$ |
| Wu-Wass | $0.575 \pm 0.028$ | $0.578 \pm 0.011$ | $0.750 \pm 0.021$ | $0.589 \pm 0.018$ |
| MVL | $0.548 \pm 0.047$ | $0.563 \pm 0.029$ | $0.738 \pm 0.031$ | $0.600 \pm 0.024$ |
| exp_MSE | $0.495 \pm 0.040$ | $\mathbf{0.528 \pm 0.026}$ | $\mathbf{0.788 \pm 0.019}$ | $0.628 \pm 0.025$ |

**Table C.5.** Results on the IDRiD dataset. Performance in terms of the Average Mean Absolute Error (AMAE) [32], Uniform Ordinal Classification Index ($A_{\mathrm{UOC}}$) [33], Kendall's $\tau_b$ [34] and the balanced accuracy, comparing: Cross-Entropy (CE), Ordinal Encoding (OE) [14], Binomial Cross-Entropy (BCE) [5], UnimodalNet (UN) [7], $L_p$ [6], Wasserstein-Unimodal-Wasserstein (Wu-Wass) [7], and Mean-Variance Loss (MVL) [25] to our proposed method exp_MSE. Arrows indicate the direction of better performance for each metric. The best results are reported in **bold**, the second best results are underlined.

| Experiment | AMAE ↓ | $A_{\mathrm{UOC}}$ ↓ | Kendall's $\tau_b$ ↑ | BA ↑ |
|---|---|---|---|---|
| CE | $0.791 \pm 0.045$ | $0.714 \pm 0.019$ | $\mathbf{0.606 \pm 0.040}$ | $0.423 \pm 0.025$ |
| OE | $0.814 \pm 0.045$ | $0.723 \pm 0.027$ | $0.577 \pm 0.030$ | $0.412 \pm 0.039$ |
| BCE | $0.876 \pm 0.063$ | $0.746 \pm 0.027$ | $0.543 \pm 0.062$ | $0.383 \pm 0.041$ |
| UN | $0.919 \pm 0.048$ | $0.741 \pm 0.024$ | $0.594 \pm 0.053$ | $0.391 \pm 0.031$ |
| $L_p$ | $0.879 \pm 0.045$ | $0.755 \pm 0.010$ | $0.568 \pm 0.016$ | $0.367 \pm 0.017$ |
| Wu-Wass | $\mathbf{0.777 \pm 0.040}$ | $\mathbf{0.711 \pm 0.014}$ | $0.602 \pm 0.026$ | $\mathbf{0.424 \pm 0.031}$ |
| MVL | $0.856 \pm 0.041$ | $0.733 \pm 0.024$ | $0.562 \pm 0.041$ | $0.414 \pm 0.047$ |
| exp_MSE | $0.787 \pm 0.018$ | $0.717 \pm 0.009$ | $0.580 \pm 0.019$ | $0.419 \pm 0.017$ |

## C.2 Confusion Matrices and Class Distributions

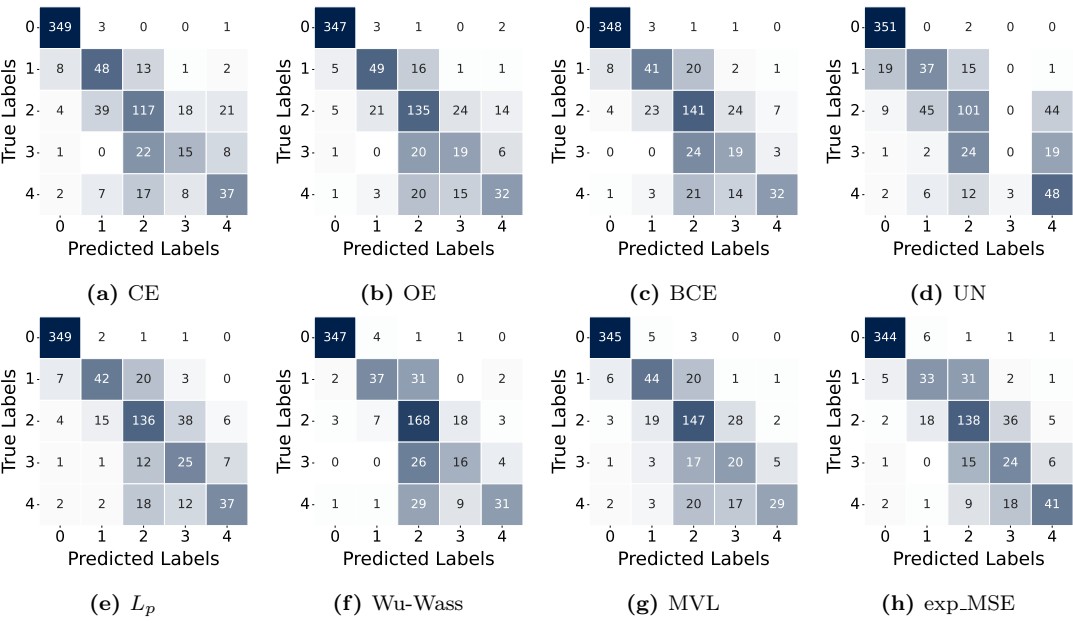

**Figure C.1.** Confusion matrices on the Aptos dataset for train/validation split seed 0 comparing: Cross-Entropy (CE), Ordinal Encoding (OE) [14], Binomial Cross-Entropy (BCE) [5], UnimodalNet (UN) [7], $L_p$ [6], Wasserstein-Unimodal-Wasserstein (Wu-Wass) [7], and Mean-Variance Loss (MVL) [25] to our proposed method exp_MSE. The color intensity is normalized per row.

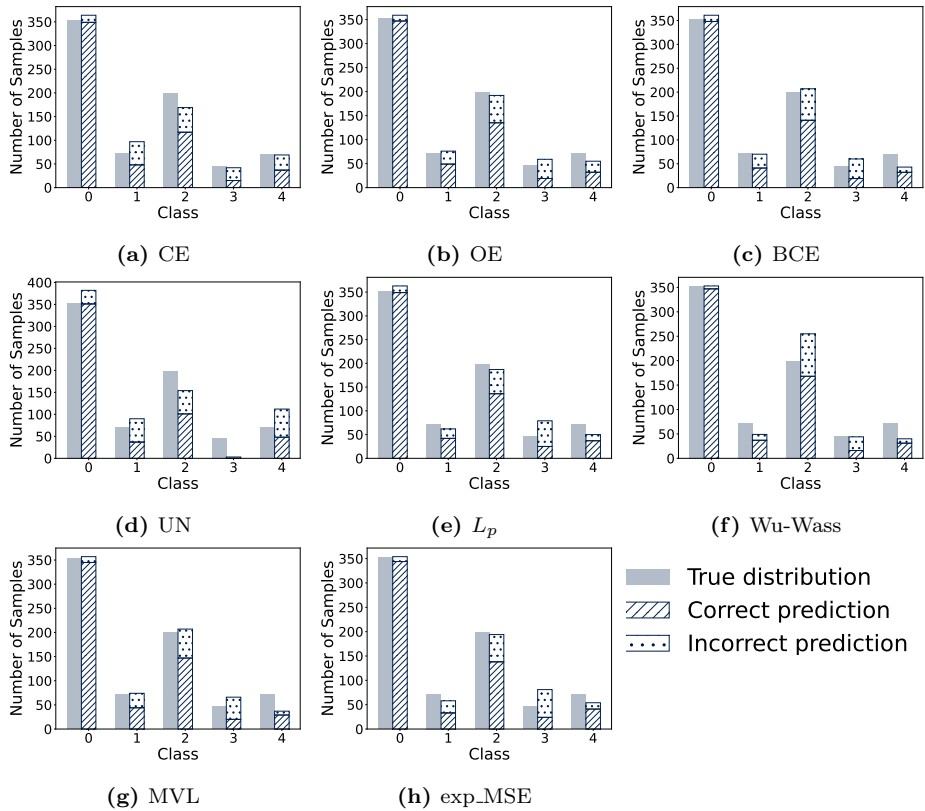

**Figure C.2.** True and predicted data distributions of the Aptos dataset for train/validation split seed 0 comparing: Cross-Entropy (CE), Ordinal Encoding (OE) [14], Binomial Cross-Entropy (BCE) [5], UnimodalNet (UN) [7], $L_p$ [6], Wasserstein-Unimodal-Wasserstein (Wu-Wass) [7], and Mean-Variance Loss (MVL) [25] to our proposed method exp_MSE. Predicted data distributions distinguish between correct and incorrect predictions.

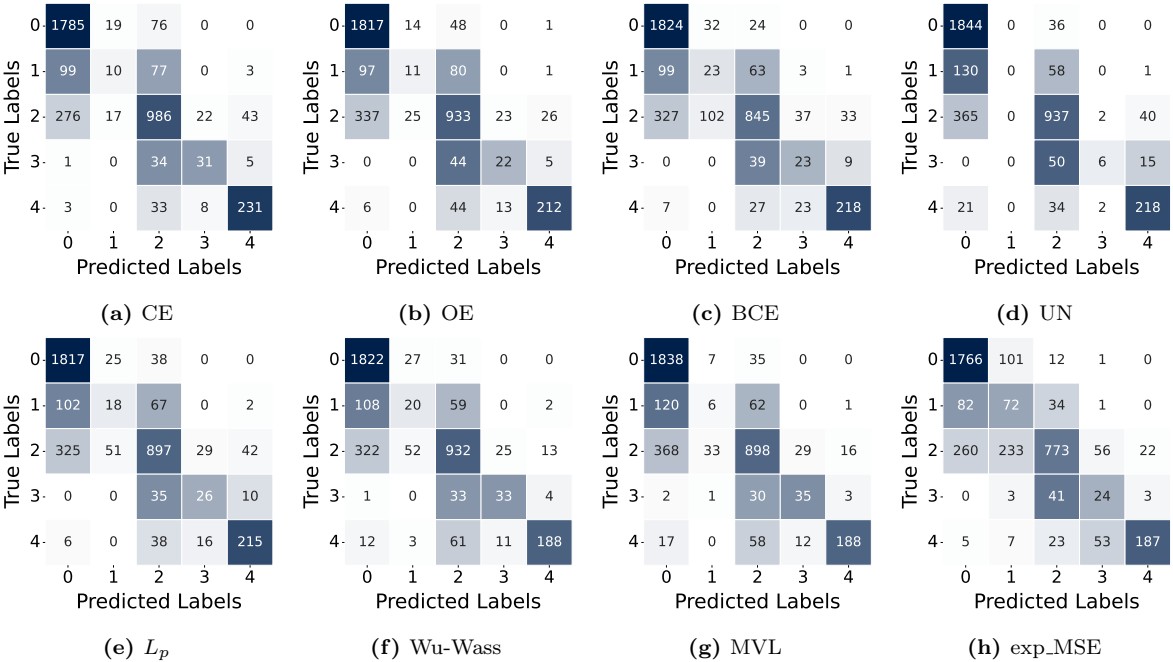

**Figure C.3.** Confusion matrices on the DDR dataset for train/validation split seed 0 comparing: Cross-Entropy (CE), Ordinal Encoding (OE) [14], Binomial Cross-Entropy (BCE) [5], UnimodalNet (UN) [7], $L_p$ [6], Wasserstein-Unimodal-Wasserstein (Wu-Wass) [7], and Mean-Variance Loss (MVL) [25] to our proposed method exp_MSE. The color intensity is normalized per row.

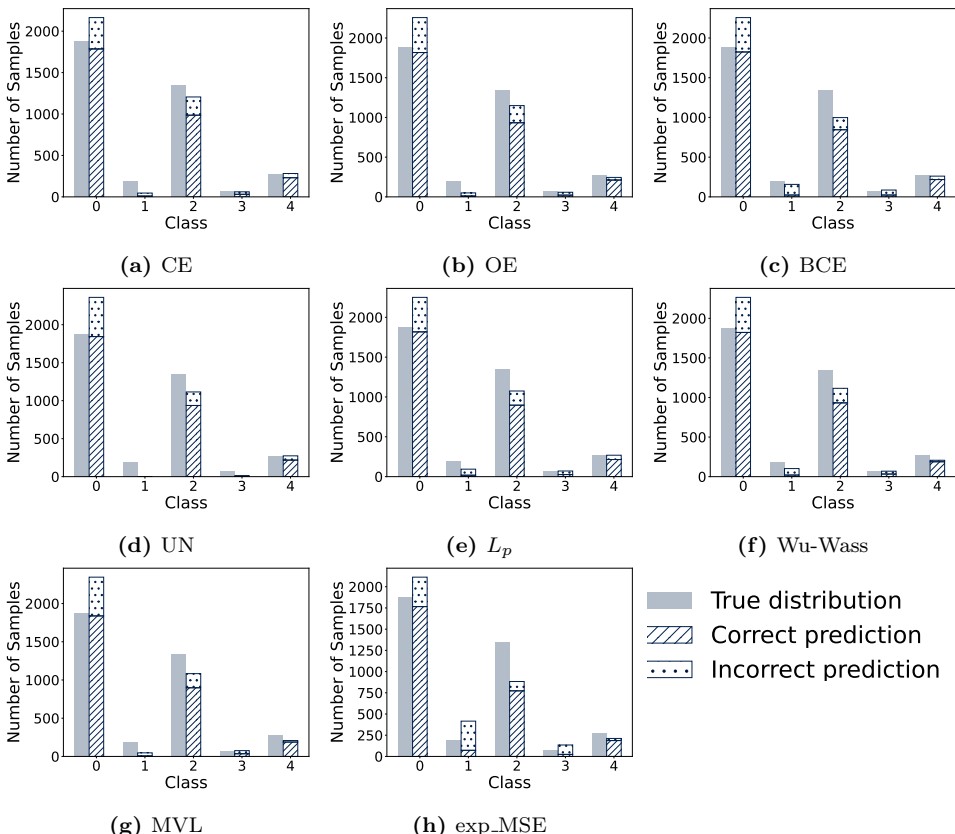

**Figure C.4.** True and predicted data distributions of the DDR dataset for train/validation split seed 0 comparing: Cross-Entropy (CE), Ordinal Encoding (OE) [14], Binomial Cross-Entropy (BCE) [5], UnimodalNet (UN) [7], $L_p$ [6], Wasserstein-Unimodal-Wasserstein (Wu-Wass) [7], and Mean-Variance Loss (MVL) [25] to our proposed method exp_MSE. Predicted data distributions distinguish between correct and incorrect predictions.

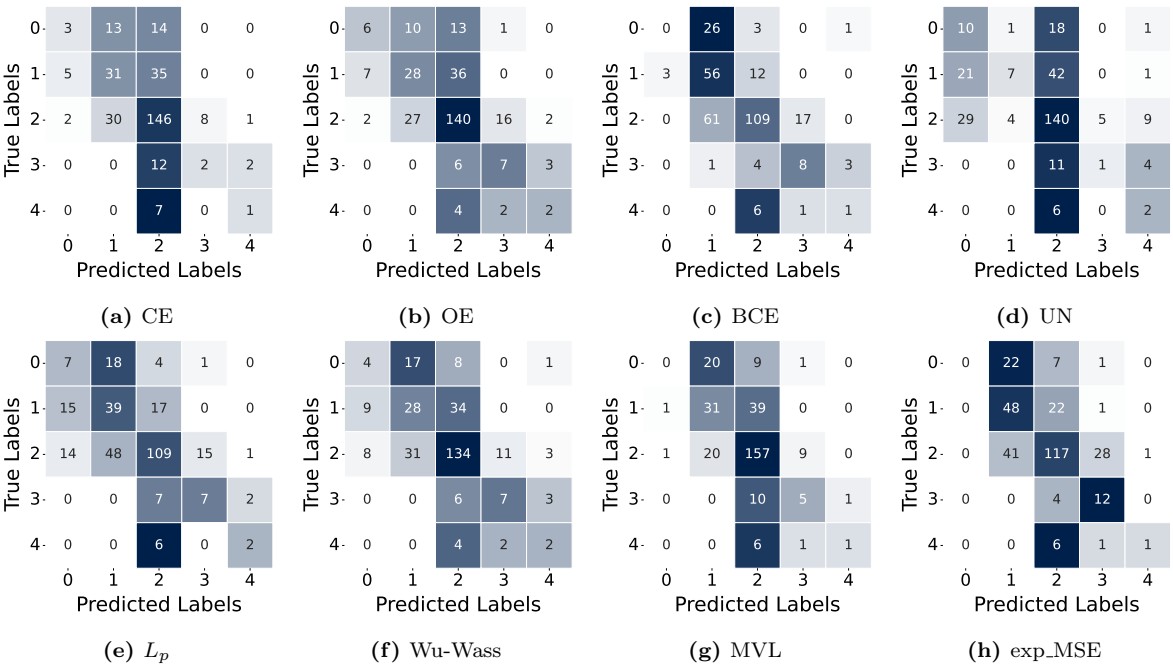

**Figure C.5.** Confusion matrices on the RLDR dataset for train/validation split seed 0 comparing: Cross-Entropy (CE), Ordinal Encoding (OE) [14], Binomial Cross-Entropy (BCE) [5], UnimodalNet (UN) [7], $L_p$ [6], Wasserstein-Unimodal-Wasserstein (Wu-Wass) [7], and Mean-Variance Loss (MVL) [25] to our proposed method exp_MSE. The color intensity is normalized per row.

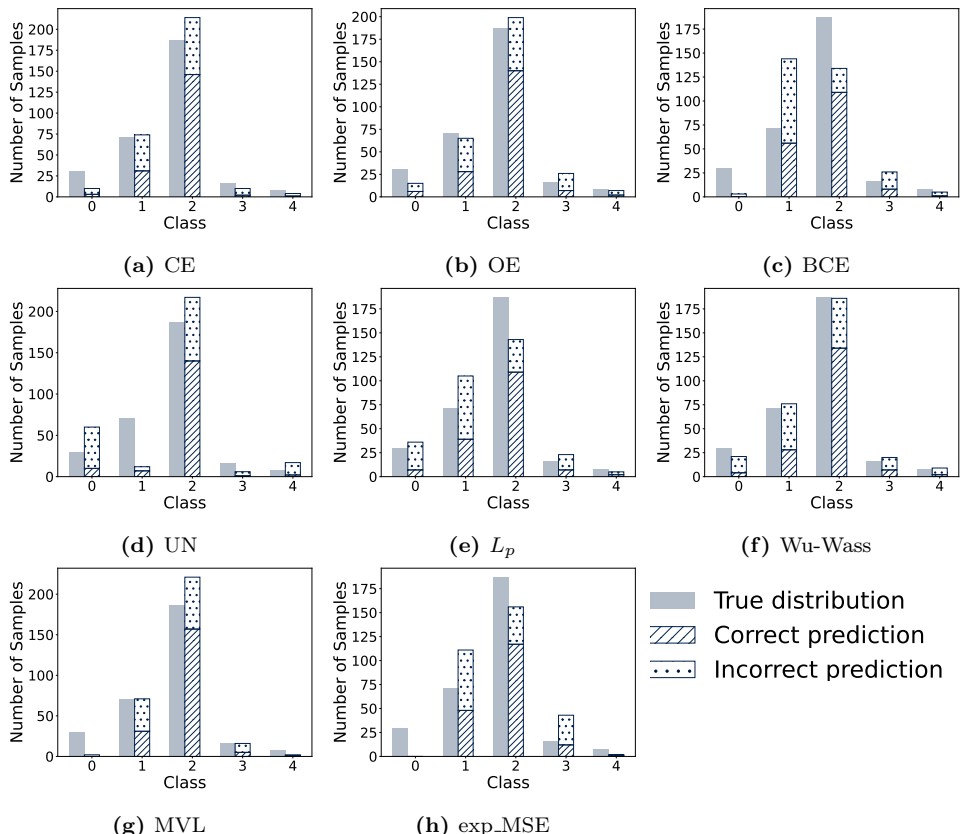

**Figure C.6.** True and predicted data distributions of the RLDR dataset for train/validation split seed 0 comparing: Cross-Entropy (CE), Ordinal Encoding (OE) [14], Binomial Cross-Entropy (BCE) [5], UnimodalNet (UN) [7], $L_p$ [6], Wasserstein-Unimodal-Wasserstein (Wu-Wass) [7], and Mean-Variance Loss (MVL) [25] to our proposed method exp_MSE. Predicted data distributions distinguish between correct and incorrect predictions.

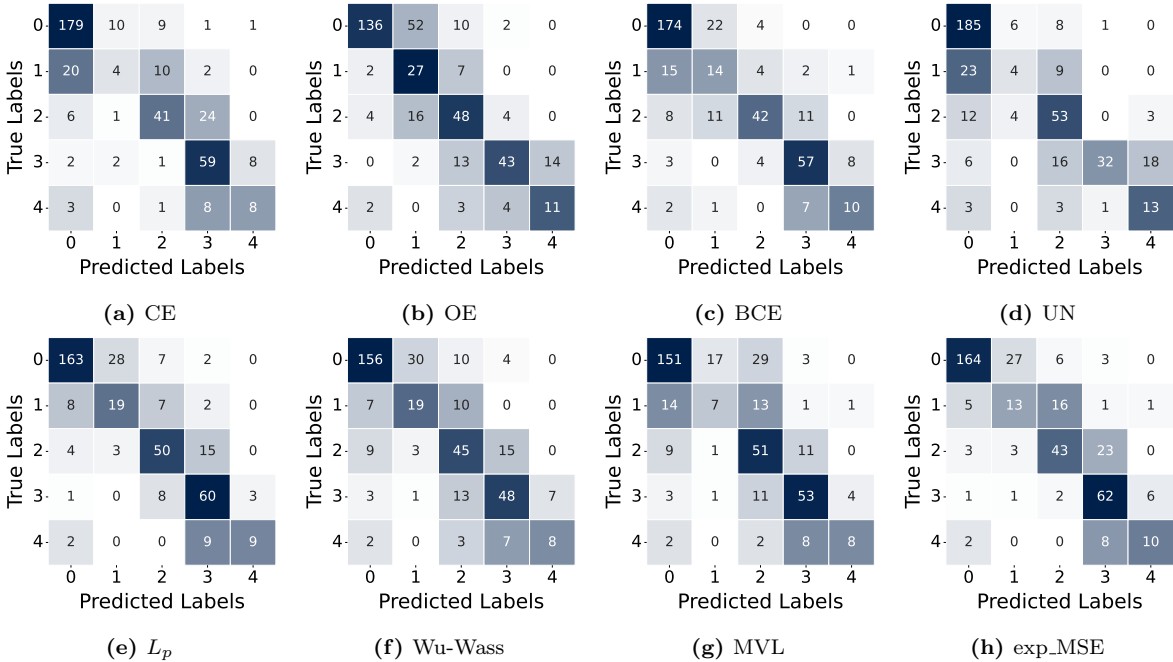

**Figure C.7.** Confusion matrices on the DeepDR dataset for train/validation split seed 0 comparing: Cross-Entropy (CE), Ordinal Encoding (OE) [14], Binomial Cross-Entropy (BCE) [5], UnimodalNet (UN) [7], $L_p$ [6], Wasserstein-Unimodal-Wasserstein (Wu-Wass) [7], and Mean-Variance Loss (MVL) [25] to our proposed method exp_MSE. The color intensity is normalized per row.

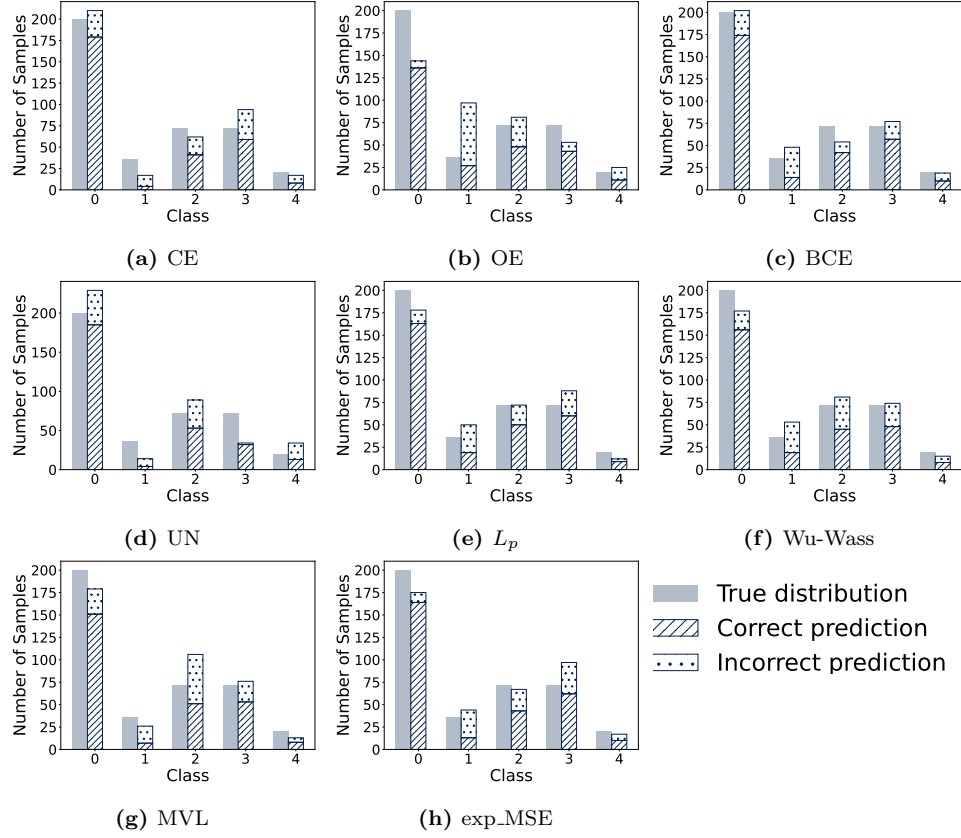

**Figure C.8.** True and predicted data distributions of the DeepDR dataset for train/validation split seed 0 comparing: Cross-Entropy (CE), Ordinal Encoding (OE) [14], Binomial Cross-Entropy (BCE) [5], UnimodalNet (UN) [7], $L_p$ [6], Wasserstein-Unimodal-Wasserstein (Wu-Wass) [7], and Mean-Variance Loss (MVL) [25] to our proposed method exp_MSE. Predicted data distributions distinguish between correct and incorrect predictions.

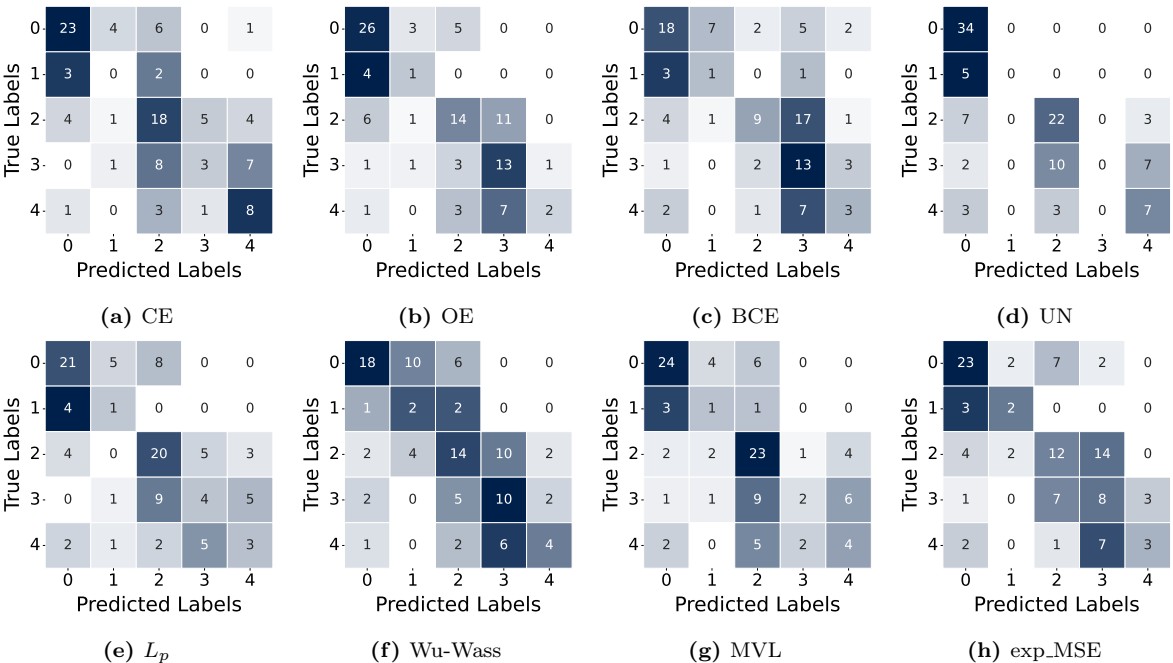

**Figure C.9.** Confusion matrices on the IDRiD dataset for train/validation split seed 0 comparing: Cross-Entropy (CE), Ordinal Encoding (OE) [14], Binomial Cross-Entropy (BCE) [5], UnimodalNet (UN) [7], $L_p$ [6], Wasserstein-Unimodal-Wasserstein (Wu-Wass) [7], and Mean-Variance Loss (MVL) [25] to our proposed method exp_MSE. The color intensity is normalized per row.

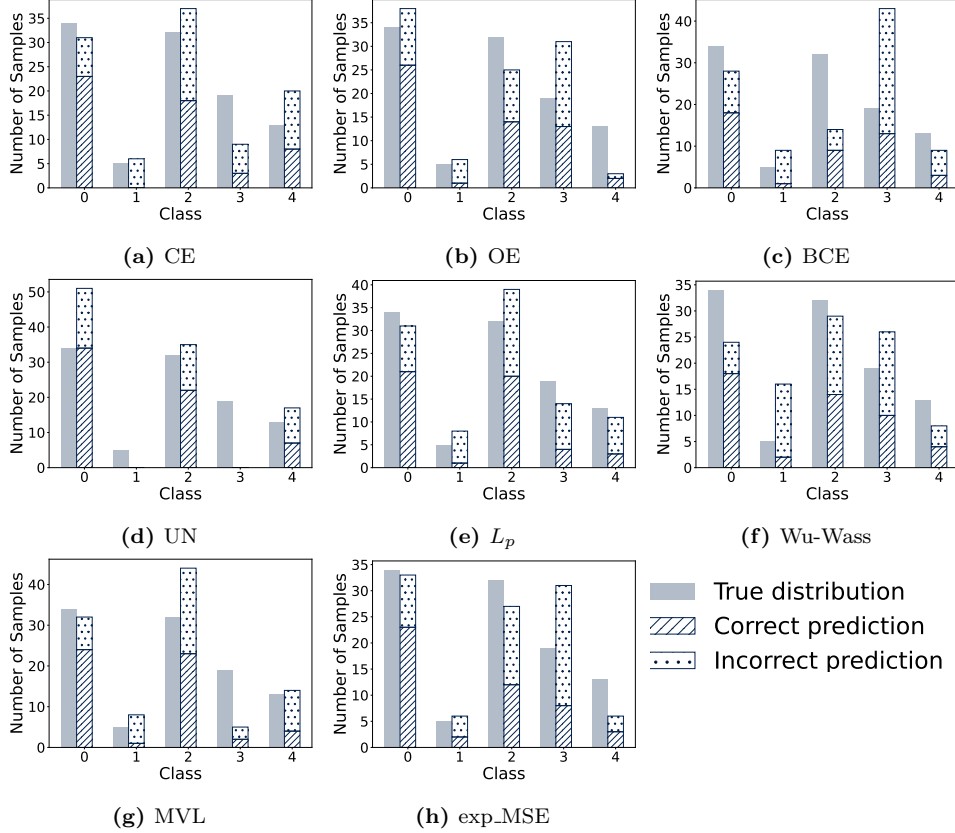

**Figure C.10.** True and predicted data distributions of the IDRiD dataset for train/validation split seed 0 comparing: Cross-Entropy (CE), Ordinal Encoding (OE) [14], Binomial Cross-Entropy (BCE) [5], UnimodalNet (UN) [7], $L_p$ [6], Wasserstein-Unimodal-Wasserstein (Wu-Wass) [7], and Mean-Variance Loss (MVL) [25] to our proposed method exp_MSE. Predicted data distributions distinguish between correct and incorrect predictions.

## C.3 Feature Analysis

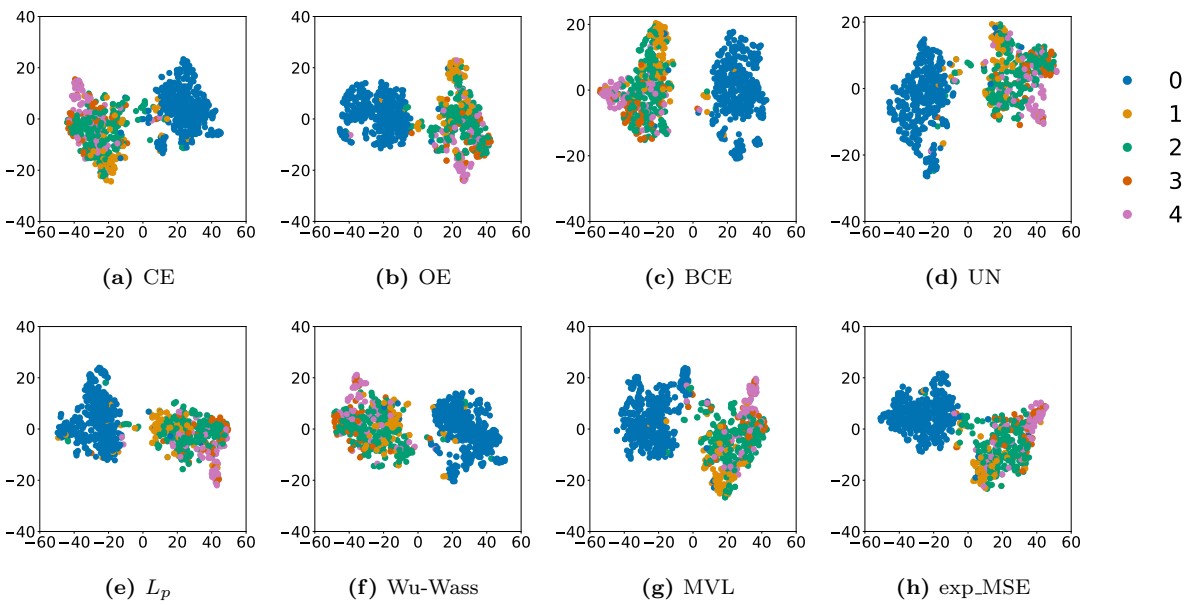

**(a)** CE     **(b)** OE     **(c)** BCE     **(d)** UN

**(e)** $L_p$     **(f)** Wu-Wass     **(g)** MVL     **(h)** exp_MSE

**Figure C.11.** t-SNE [40] analysis on the Aptos dataset for train/validation split seed 0 comparing: Cross-Entropy (CE), Ordinal Encoding (OE) [14], Binomial Cross-Entropy (BCE) [5], UnimodalNet (UN) [7], $L_p$ [6], Wasserstein-Unimodal-Wasserstein (Wu-Wass) [7], and Mean-Variance Loss (MVL) [25] to our proposed method exp_MSE.

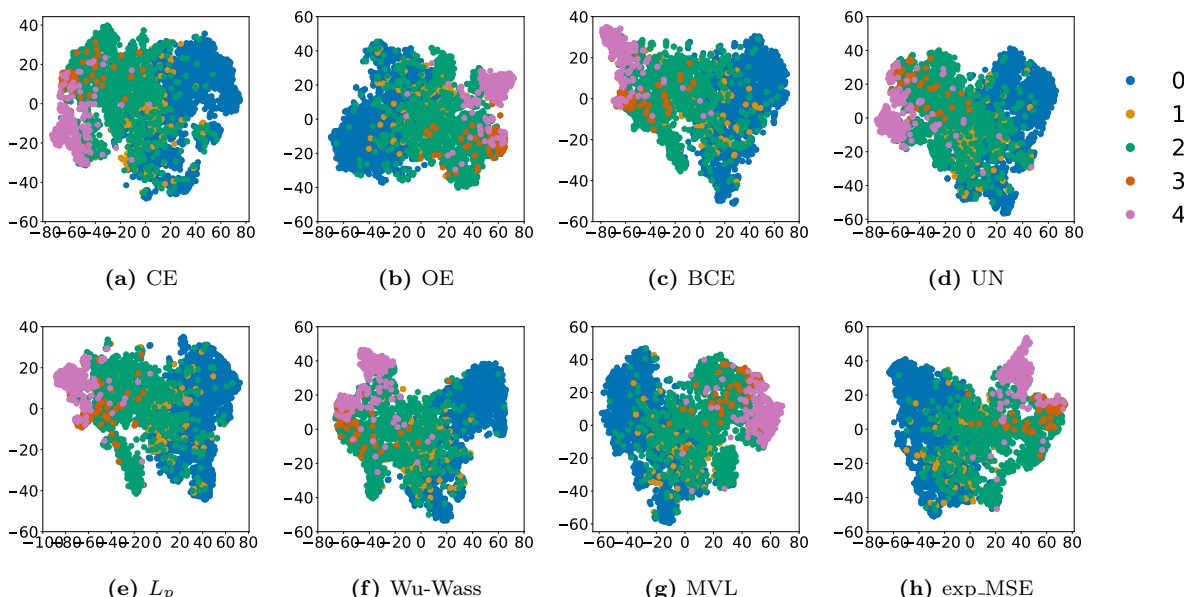

**(a)** CE     **(b)** OE     **(c)** BCE     **(d)** UN

**(e)** $L_p$     **(f)** Wu-Wass     **(g)** MVL     **(h)** exp_MSE

**Figure C.12.** t-SNE [40] analysis on the DDR dataset for train/validation split seed 0 comparing: Cross-Entropy (CE), Ordinal Encoding (OE) [14], Binomial Cross-Entropy (BCE) [5], UnimodalNet (UN) [7], $L_p$ [6], Wasserstein-Unimodal-Wasserstein (Wu-Wass) [7], and Mean-Variance Loss (MVL) [25] to our proposed method exp_MSE.

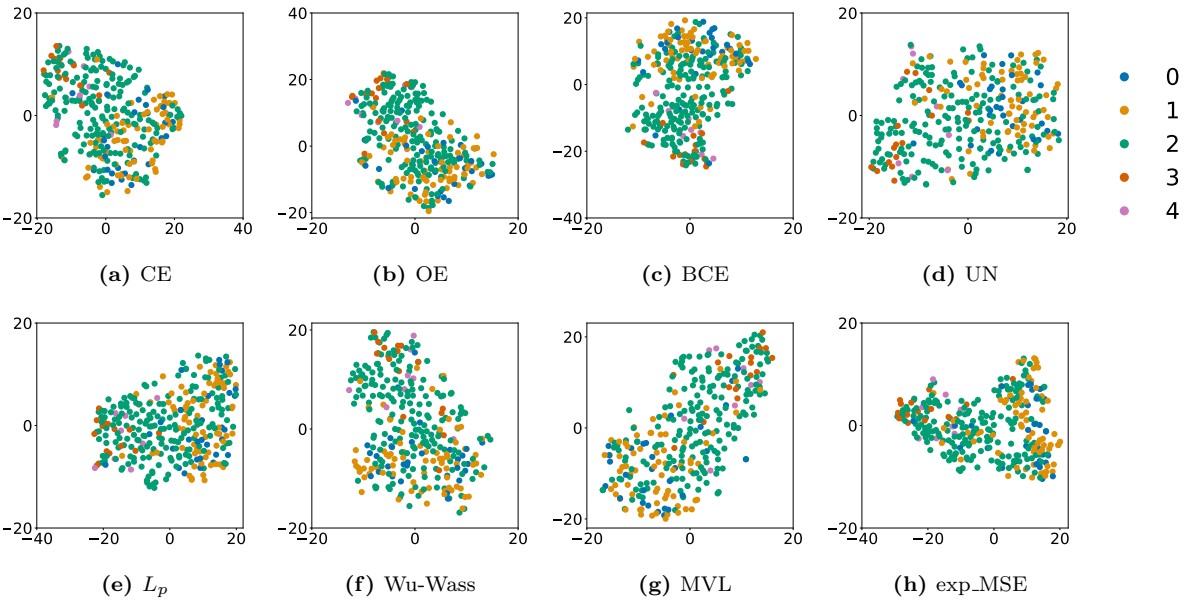

**Figure C.13.** t-SNE [40] analysis on the RLDR dataset for train/validation split seed 0 comparing: Cross-Entropy (CE), Ordinal Encoding (OE) [14], Binomial Cross-Entropy (BCE) [5], UnimodalNet (UN) [7], $L_p$ [6], Wasserstein-Unimodal-Wasserstein (Wu-Wass) [7], and Mean-Variance Loss (MVL) [25] to our proposed method exp_MSE.

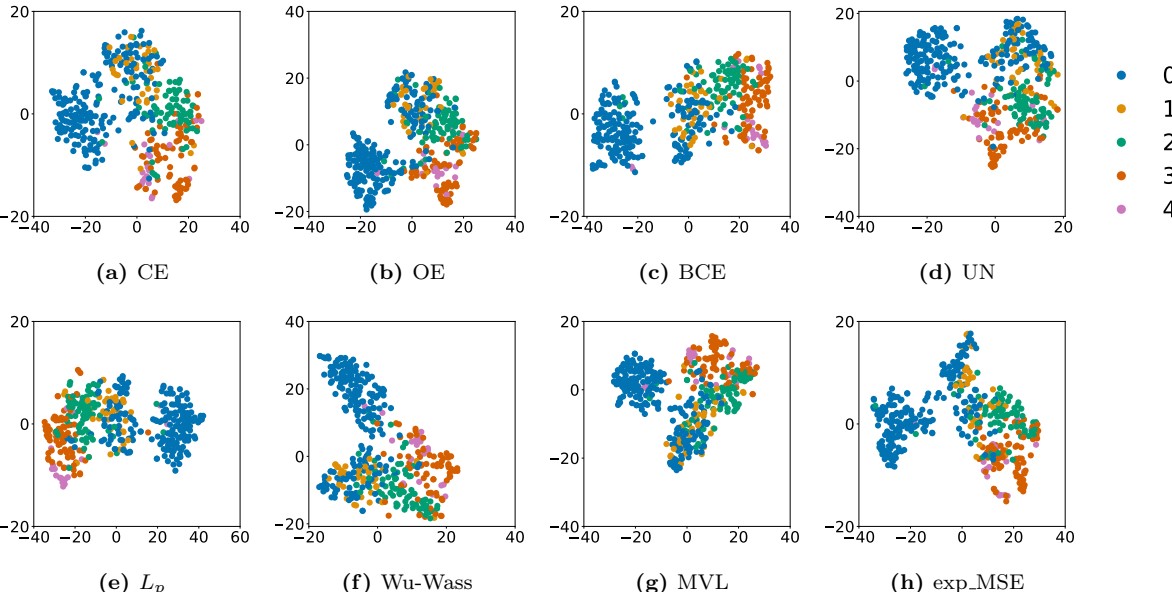

**Figure C.14.** t-SNE [40] analysis on the DeepDR dataset for train/validation split seed 0 comparing: Cross-Entropy (CE), Ordinal Encoding (OE) [14], Binomial Cross-Entropy (BCE) [5], UnimodalNet (UN) [7], $L_p$ [6], Wasserstein-Unimodal-Wasserstein (Wu-Wass) [7], and Mean-Variance Loss (MVL) [25] to our proposed method exp_MSE.

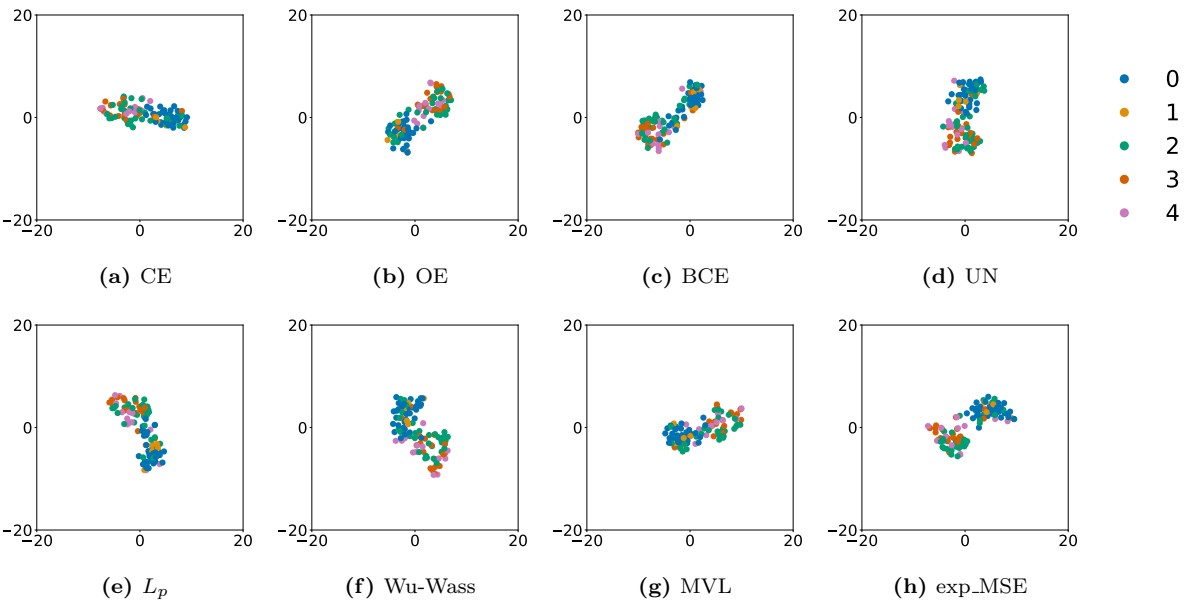

**Figure C.15.** t-SNE [40] analysis on the IDRiD dataset for train/validation split seed 0 comparing: Cross-Entropy (CE), Ordinal Encoding (OE) [14], Binomial Cross-Entropy (BCE) [5], UnimodalNet (UN) [7], $L_p$ [6], Wasserstein-Unimodal-Wasserstein (Wu-Wass) [7], and Mean-Variance Loss (MVL) [25] to our proposed method exp_MSE.

## C.4 Posterior Distributions

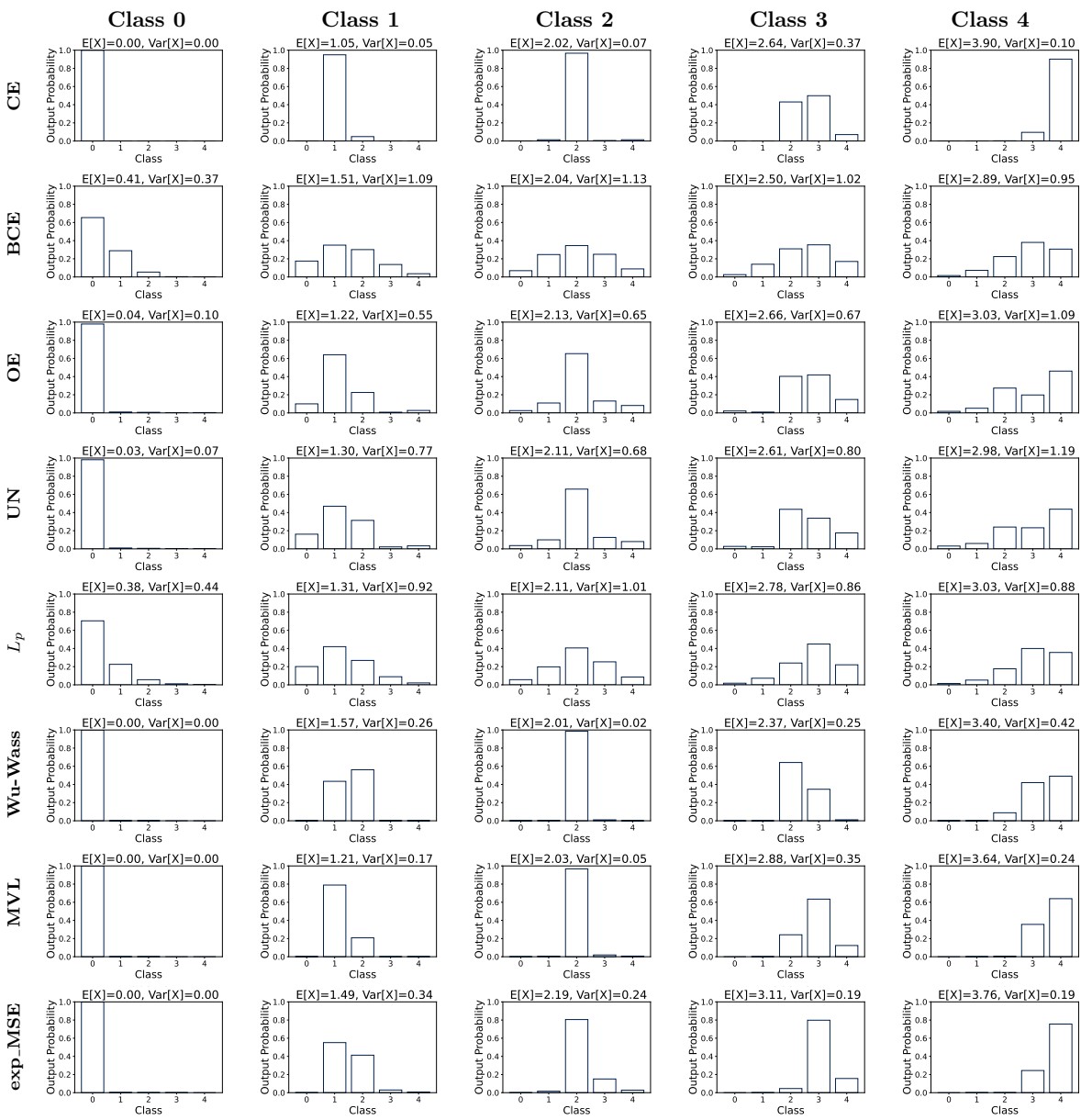

**Figure C.16.** Normalized posterior distributions on the Aptos dataset for the train/validation split seed 0, shown per true class (columns) comparing the methods (rows): Cross-Entropy (CE), Ordinal Encoding (OE) [14], Binomial Cross-Entropy (BCE) [5], UnimodalNet (UN) [7], $L_p$ [6], Wasserstein-Unimodal-Wasserstein (Wu-Wass) [7], and Mean-Variance Loss (MVL) [25] to our proposed method exp_MSE.

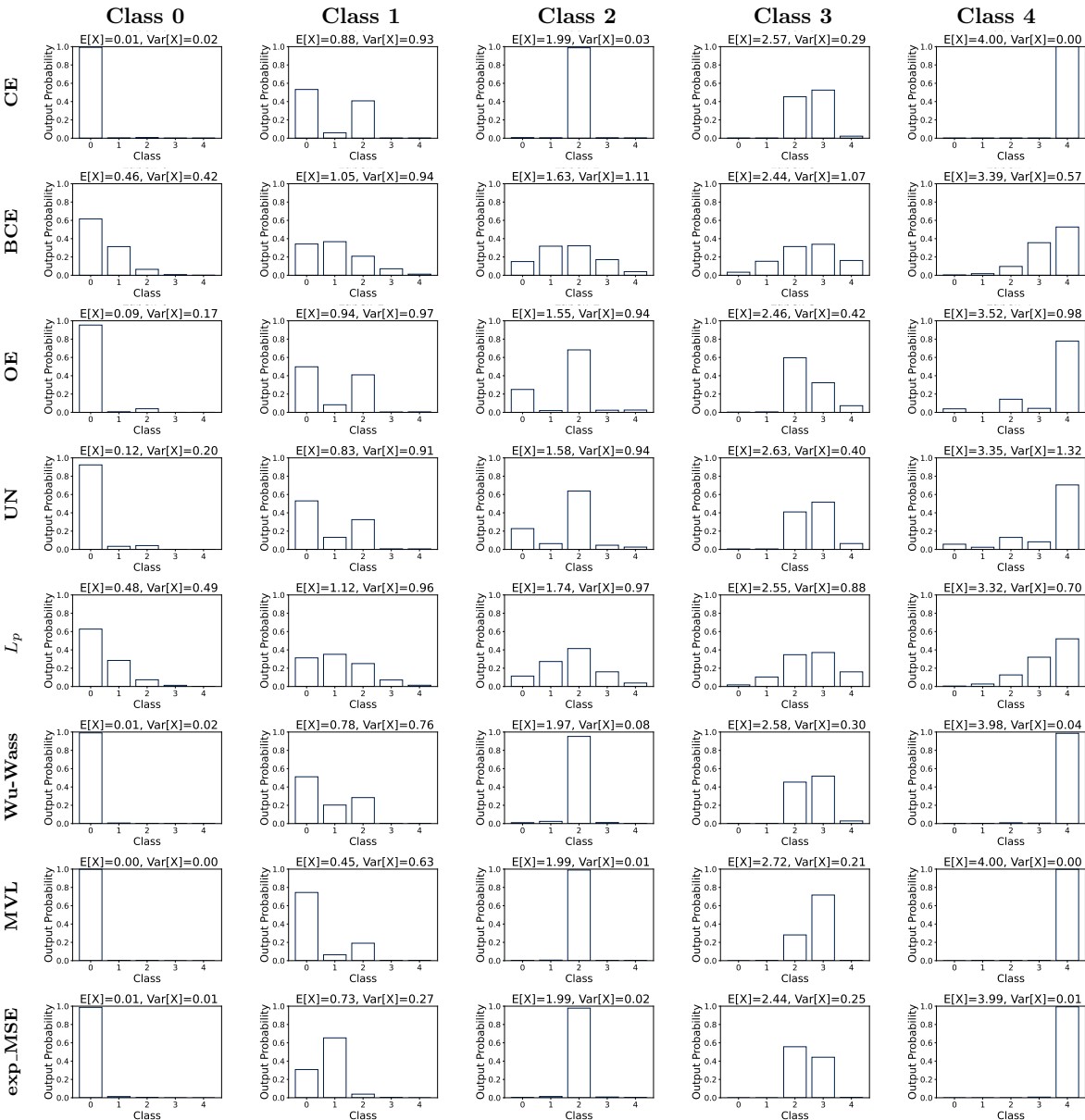

**Figure C.17.** Normalized posterior distributions on the DDR dataset for the train/validation split seed 0, shown per true class (columns) comparing the methods (rows): Cross-Entropy (CE), Ordinal Encoding (OE) [14], Binomial Cross-Entropy (BCE) [5], UnimodalNet (UN) [7], $L_p$ [6], Wasserstein-Unimodal-Wasserstein (Wu-Wass) [7], and Mean-Variance Loss (MVL) [25] to our proposed method exp_MSE.

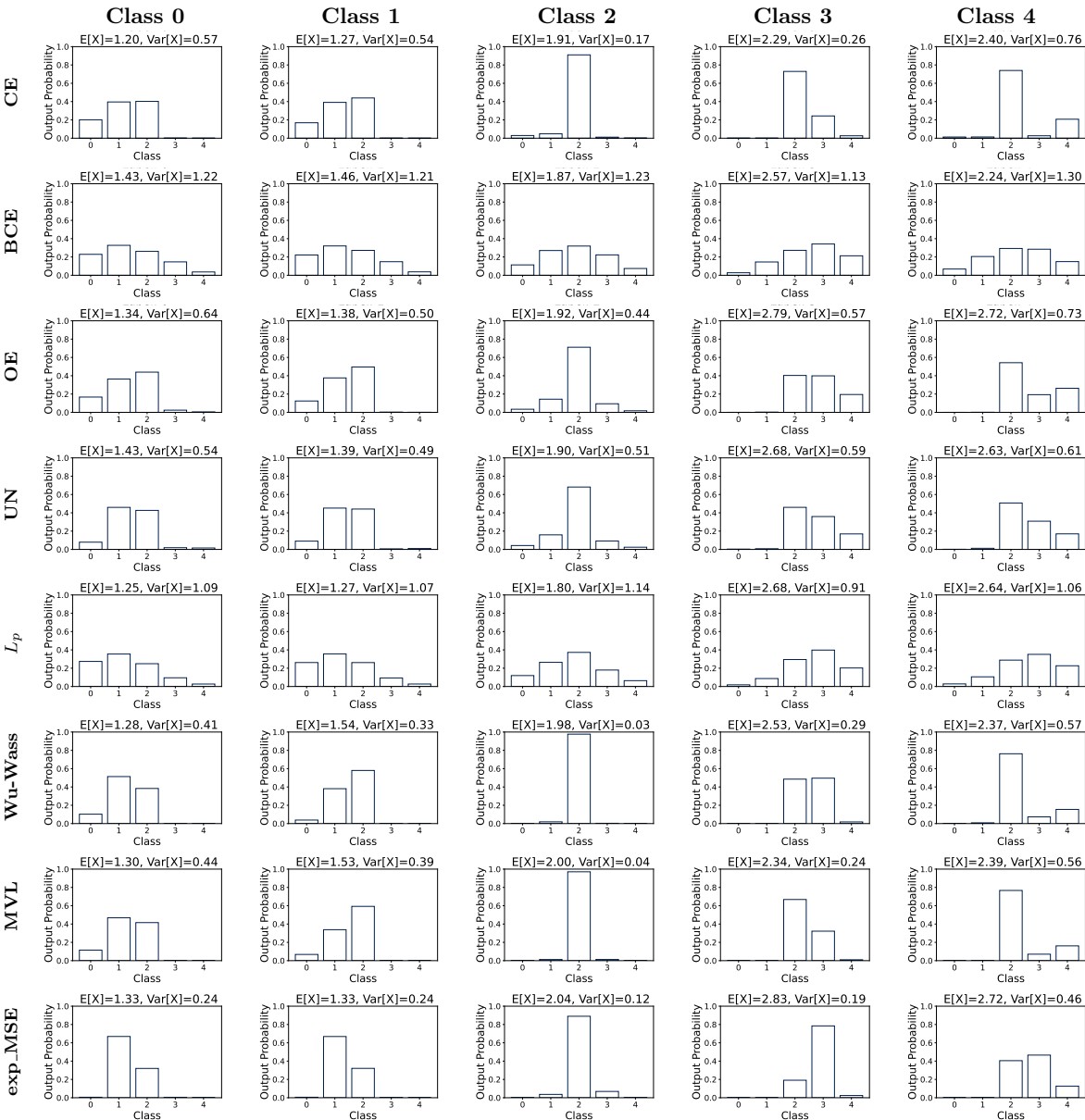

**Figure C.18.** Normalized posterior distributions on the RLDR dataset for the train/validation split seed 0, shown per true class (columns) comparing the methods (rows): Cross-Entropy (CE), Ordinal Encoding (OE) [14], Binomial Cross-Entropy (BCE) [5], UnimodalNet (UN) [7], $L_p$ [6], Wasserstein-Unimodal-Wasserstein (Wu-Wass) [7], and Mean-Variance Loss (MVL) [25] to our proposed method exp_MSE.

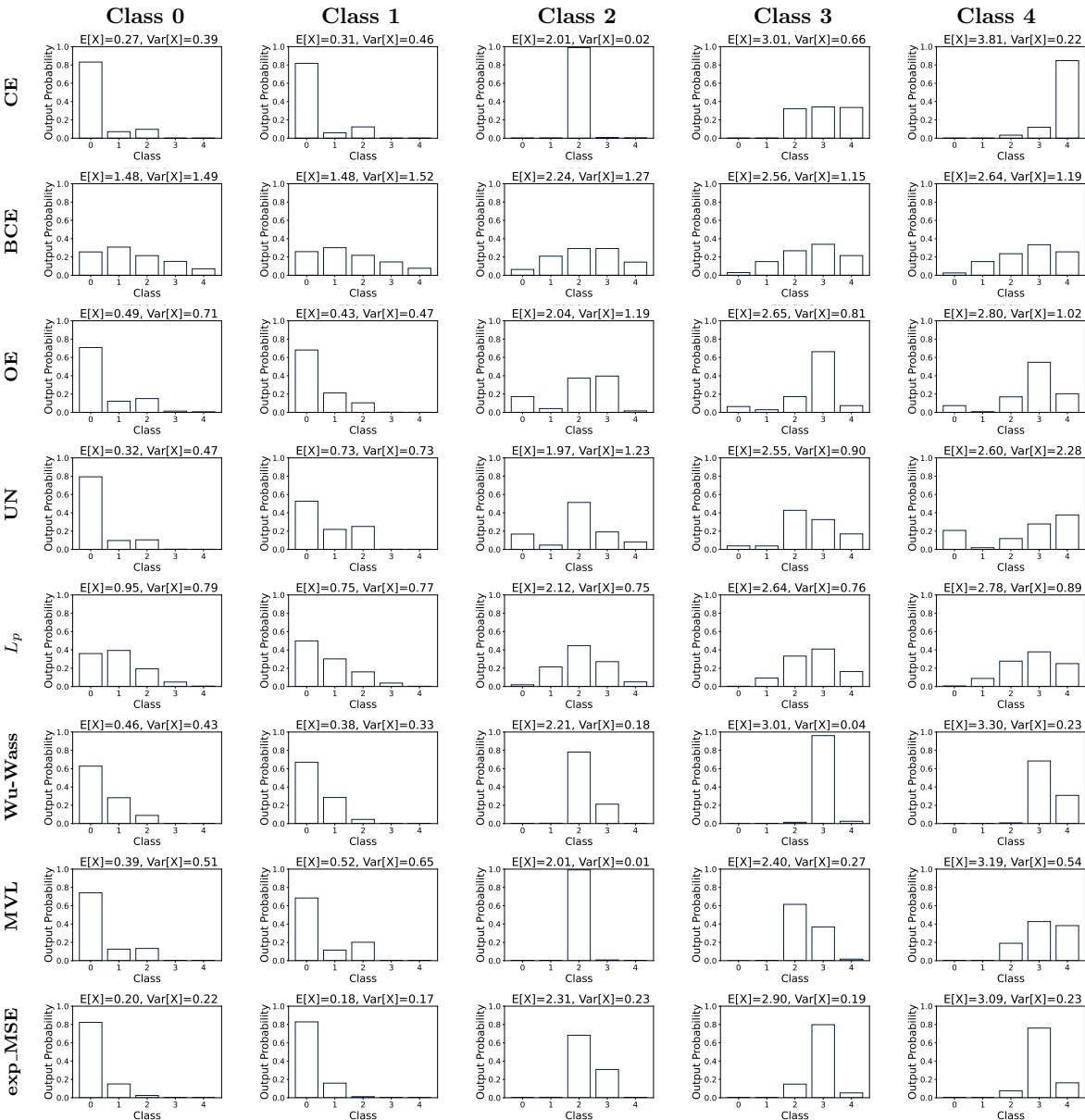

**Figure C.19.** Normalized posterior distributions on the IDRiD dataset for the train/validation split seed 0, shown per true class (columns) comparing the methods (rows): Cross-Entropy (CE), Ordinal Encoding (OE) [14], Binomial Cross-Entropy (BCE) [5], UnimodalNet (UN) [7], $L_p$ [6], Wasserstein-Unimodal-Wasserstein (Wu-Wass) [7], and Mean-Variance Loss (MVL) [25] to our proposed method exp_MSE.

