# OpenReview forum: "Preserving Ordinality in Diabetic Retinopathy Grading through a Distribution-Based Loss Function"
_NLDL.org/2026/Conference — NLDL 2026 Poster_

### Official Review · Reviewer_g3yL · 2025-10-02
**The review of "Preserving Ordinality in Diabetic Retinopathy Grading through a Distribution-Based Loss Function"**

**Rating:** 2
**Confidence:** 4

**Summary:**

This paper introduces a loss function, Expectation Mean Squared Error (exp MSE), for the task of ordinal classification in Diabetic Retinopathy (DR) grading. The method operates by penalizing the mean and variance of the model's output probability distribution to encourage accurate and concentrated predictions. Experiments show that this method has certain effectiveness.

**Strengths:**

1. The paper's claims are supported by experiments across five public DR datasets, which lends significant weight to the empirical findings.
2. The authors compare their method against a well-chosen spectrum of baselines, including nominal (Cross-Entropy), soft-unimodal (BCE, Lp), and hard-unimodal (UnimodalNet) approaches, providing a solid context for their results.

**Weaknesses:**

1. Exp MSE is not a novel loss function. Mathematically, it is merely a special case of the established Mean-Variance Loss (MVL), obtained by removing the cross-entropy component. Therefore, it does not represent a fundamental contribution.

2. The paper lacks an ablation study to justify why simplifying MVL by discarding the cross-entropy term leads to improved performance. This omission misses a valuable opportunity for deeper insight. Without such analysis, the claimed contribution remains an empirical observation rather than a well-substantiated finding.

3. The paper attributes the model’s robustness primarily to Exp MSE. However, this claim is confounded by the use of a strong data augmentation strategy in the experimental setup. The observed performance gains are more likely the result of a synergistic effect between the loss function and data augmentation. Yet, the paper does not provide the necessary ablation studies to disentangle and clarify their individual contributions.

4. The paper omits several important references, particularly notable works on ranking losses that are closely related to this topic [1,2,3,4].

[1] LibAUC: A Deep Learning Library for X-risk Optimization. KDD 2023.

[2] When all we need is a piece of the pie: A generic framework for optimizing two-way partial AUC. ICML 2021.

[3] Large-scale robust deep auc maximization: A new surrogate loss and empirical studies on medical image classification. ICCV 2021.

[4] Aucseg: Auc-oriented pixel-level long-tail semantic segmentation. NeurIPS 2024.

**Justification:**

My recommendation for rejection is based on the fact that the authors did not clearly articulate the main contributions of the paper and omitted several important references. In addition, the paper lacks the necessary ablation studies. That said, the proposed method and experimental results are still commendable. If the authors could address these issues with targeted improvements, I would be willing to raise my score.

---

> ### Author Rebuttal · Authors · 2025-10-22
>
> We thank the reviewer for their valuable and insightful feedback.
>
>
> **General note:**  After submission, we realized that some reported results were incorrect. We have verified and corrected them for the camera-ready version if the paper is accepted. These updates, which affect Wu-Wass, MVL, and exp_MSE, do not change the conclusions of our paper, rather strengthen our claims. The updated results (blue emoji) are:
>
> | Experiment | AMAE ↓ | A_UOC ↓ | Kendall's τ_b ↑ | BA ↑ |
> |------------|---------:|----------:|----------------:|-----:|
> | CE         | 0.668 ± 0.060 | 0.631 ± 0.031 | 0.674 ± 0.025 | 0.518 ± 0.036 |
> | OE         | 0.653 ± 0.051 | 0.626 ± 0.029 | 0.663 ± 0.031 | 0.520 ± 0.034 |
> | BCE        | 0.659 ± 0.031 | 0.634 ± 0.018 | 0.680 ± 0.026 | 0.509 ± 0.026 |
> | UN         | 0.766 ± 0.042 | 0.675 ± 0.020 | 0.659 ± 0.033 | 0.465 ± 0.026 |
> | L_p  | *0.648 ± 0.036* | *0.622 ± 0.017* | *0.683 ± 0.017*  | *0.522 ± 0.022*  |
> | Wu-Wass    | 0.653 ± 0.035 | 0.628 ± 0.014 | 🔵 0.676 ± 0.021 | 🔵 0.520 ± 0.018 |
> | MVL        | 🔵 0.667 ± 0.034 | 🔵 0.630 ± 0.018 | 🔵 0.670 ± 0.023 | 🔵 0.518 ± 0.021 |
> | exp_MSE    | **0.616 ± 0.047** | **0.609 ± 0.027** | **0.695 ± 0.024** | 🔵 **0.535 ± 0.031** |
>
> - **Weakness 1:**
>   - *Reviewer's comment:* ‘Exp MSE is not a novel loss function. Mathematically, it is merely a special case of the established Mean-Variance Loss (MVL), obtained by removing the cross-entropy component. Therefore, it does not represent a fundamental contribution.’
>   - **Response:**  We agree with the reviewer’s concerns. The MVL was discovered close to the deadline, when our project was already finalized. Nevertheless, we believe our method demonstrates novelty, as our proposed loss achieves better results than MVL. Due to time constraints, we could not explore connections between both losses. The suggested ablations and evaluations are directions we plan to pursue in future work.
>
> - **Weakness 2:**
>   - *Reviewer's comment:* ‘The paper lacks an ablation study to justify why simplifying MVL by discarding the cross-entropy term leads to improved performance. This omission misses a valuable opportunity for deeper insight. Without such analysis, the claimed contribution remains an empirical observation rather than a well-substantiated finding.’
>   - **Response:** We agree with the reviewer’s concerns. This analysis could not be conducted as we discovered the MVL very close to the submission deadline. We plan to explore these ablations and their effects more thoroughly in future work as part of our upcoming projects.
>
> - **Weakness 3:**
>   - *Reviewer's comment:* ‘The paper attributes the model’s robustness primarily to Exp MSE. However, this claim is confounded by the use of a strong data augmentation strategy in the experimental setup. The observed performance gains are more likely the result of a synergistic effect between the loss function and data augmentation. Yet, the paper does not provide the necessary ablation studies to disentangle and clarify their individual contributions.’
>   - **Response:** We understand this concern. We used the same experimental settings for all datasets to ensure fair comparisons. The data augmentation choices were based on preliminary experiments with the cross-entropy loss. In future work, we plan to explore the impact of different data augmentation strategies on all ordinal methods.
>
> - **Weakness 4:**
>   - *Reviewer's comment:* ‘The paper omits several important references, particularly notable works on ranking losses that are closely related to this topic [1,2,3,4].’
>   - **Response:** We thank the reviewer for suggesting additional ranking losses. We are already considering seven benchmarking methods, which are better aligned with the problem we aim to solve than the losses in [1,2,3,4]. Due to time and computational constraints, we limited our evaluation to these methods.

---

### Official Review · Reviewer_Gm2L · 2025-10-03
**An interesting approach to the common problem of ordinal classification.**

**Rating:** 5
**Confidence:** 4

**Summary:**

The authors describe a novel method to introduce unimodal output in a classification model, such that the ordinal structure of the labels is preserved in the predictions.
The problem is both actual and important, as several situations in real-world modelling (not "toy examples") comprise ordinal classifications which ML models are often neglecting or unable to tackle. The proposed method makes a small step forward in proposing solutions to this issue, which is a valuable contribution. The clear exposition of the paper will likely help other researchers expand upon these ideas, maybe improving the weak points or inspiring new ways to attack this problem.

**Strengths:**

The paper is well written and clear in most of its parts. The tests are extensive. The authors did a good job at testing for different datasets, different augmentation techniques, and tested with different metrics.

**Weaknesses:**

I provide here some comments that I hope will help the authors produce a better work for publication:
- Figure 2 is unclear. It is not explained what the prediction in blue is. Is it the real class or something else? Is should be made clearer what the distribution (drawn in a continuous black line) represents. Is that the real probability distribution of the label, or the predicted distribution of the model? From the notation used for the expected value shown, it can be inferred that it is the distribution of the predicted class. If that is so, it should be made clearer. And assuming it is the distribution of the predicted class, then the class probabilities should follow it such as they do in 2b, but this does not happen in 2a. It is unclear why this would be a normal behaviour imputable to a cross-entropy (or similar) function, and not just a case of bad training. Further, the caption says that E[\hat{y} | x] is the red arrow, but that is misleading; that expected value corresponds to the highest point in the distribution (the vertical dashed line). I think this picutre should be explained better, both in the caption and in the body of the text, since it sets the stage and intuition for the whole rest of the understanding of the paper.

- In line 247 the notation y^* is used which is the true underlying posterior, but in the text it is spoken of as a prediction. Similarly, the true posterior cannot converge to anything, only a predicted posterior can converge (under training).

- The explanation of the metrics for class imbalance and data augmentation is very well written and appreciated.

- In the section "Quantitative results", I think it would be good to include a discussion on the magnitude of the differences. Yes, exp_MSE is better or second-best, but is that difference actually significative? For instance, how important, or valuable, is a difference in BA of 0.004? Or 0.037 in AMAE? etc.

- It would be great to understand a bit better why the method works well on highly-unbalanced datasets, while more uniform distributions remain more challenging. Usually it's the opposite, so that is a curious thing, which is perhaps linked to DR (of which I am no expert).

**Justification:**

I think this work makes a small step forward in tackling the difficult and recurring problem of ordinal classification. While the problem is not completely solved, the way to a solution is comprised of many small steps, and this work is one such. I am a big supporter of publishing all that helps. The authors did an extensive job at testing and supporting their claims and work, and wrote a clear and understandble paper. As such I believe this to be a valuable contribution to the field.

---

> ### Author Rebuttal · Authors · 2025-10-22
>
> We thank the reviewer for their valuable and insightful feedback.
>
>
> **General note:**  After submission, we realized that some reported results were incorrect. We have verified and corrected them for the camera-ready version if the paper is accepted. These updates, which affect Wu-Wass, MVL, and exp_MSE, do not change the conclusions of our paper, rather strengthen our claims. The updated results (blue) are:
>
> | Experiment | AMAE ↓ | A_UOC ↓ | Kendall's τ_b ↑ | BA ↑ |
> |------------|---------:|----------:|----------------:|-----:|
> | CE         | 0.668 ± 0.060 | 0.631 ± 0.031 | 0.674 ± 0.025 | 0.518 ± 0.036 |
> | OE         | 0.653 ± 0.051 | 0.626 ± 0.029 | 0.663 ± 0.031 | 0.520 ± 0.034 |
> | BCE        | 0.659 ± 0.031 | 0.634 ± 0.018 | 0.680 ± 0.026 | 0.509 ± 0.026 |
> | UN         | 0.766 ± 0.042 | 0.675 ± 0.020 | 0.659 ± 0.033 | 0.465 ± 0.026 |
> | L_p  | *0.648 ± 0.036* | *0.622 ± 0.017* | *0.683 ± 0.017*  | *0.522 ± 0.022*  |
> | Wu-Wass    | 0.653 ± 0.035 | 0.628 ± 0.014 | 🔵 0.676 ± 0.021 | 🔵 0.520 ± 0.018 |
> | MVL        | 🔵 0.667 ± 0.034 | 🔵 0.630 ± 0.018 | 🔵 0.670 ± 0.023 | 🔵 0.518 ± 0.021 |
> | exp_MSE    | **0.616 ± 0.047** | **0.609 ± 0.027** | **0.695 ± 0.024** | 🔵 **0.535 ± 0.031** |
>
>
> - **Weakness 1:**
>   - *Reviewer's comment:* ‘Figure 2 is unclear. It is not explained what the prediction in blue is. Is it the real class or something else? Is should be made clearer what the distribution (drawn in a continuous black line) represents. Is that the real probability distribution of the label, or the predicted distribution of the model? From the notation used for the expected value shown, it can be inferred that it is the distribution of the predicted class. If that is so, it should be made clearer. And assuming it is the distribution of the predicted class, then the class probabilities should follow it such as they do in 2b, but this does not happen in 2a. It is unclear why this would be a normal behaviour imputable to a cross-entropy (or similar) function, and not just a case of bad training. Further, the caption says that E[\hat{y} | x] is the red arrow, but that is misleading; that expected value corresponds to the highest point in the distribution (the vertical dashed line). I think this picutre should be explained better, both in the caption and in the body of the text, since it sets the stage and intuition for the whole rest of the understanding of the paper’
>   - **Response:** We appreciate the careful reading, and will be sure to address those points for the camera-ready version.
>
> - **Weakness 2:**
>   - *Reviewer's comment:* ‘In line 247 the notation y^* is used which is the true underlying posterior, but in the text it is spoken of as a prediction. Similarly, the true posterior cannot converge to anything, only a predicted posterior can converge (under training).’
>   - **Response:** We thank the reviewer for detecting this mistake. We will correct this in the camera-ready version.
>
> - **Weakness 4:**
>   - *Reviewer's comment:* ‘In the section "Quantitative results", I think it would be good to include a discussion on the magnitude of the differences. Yes, exp_MSE is better or second-best, but is that difference actually significative? For instance, how important, or valuable, is a difference in BA of 0.004? Or 0.037 in AMAE? etc.’
>   - **Response:** In line with common practice in the literature, we did not include statistical tests. Although the performance differences are modest, our method consistently outperformed the others on average and across most datasets.
>
> - **Weakness 5:**
>   - *Reviewer's comment:* ‘It would be great to understand a bit better why the method works well on highly-unbalanced datasets, while more uniform distributions remain more challenging. Usually it's the opposite, so that is a curious thing, which is perhaps linked to DR (of which I am no expert).’
>   - **Response:** We agree that this is an interesting topic and plan to address it in future studies.

---

### Official Review · Reviewer_1gRT · 2025-10-08
**concrete propsal**

**Rating:** 5
**Confidence:** 4
**Final Rating:** 4
**Final Confidence:** 3

**Summary:**

This paper considers a specific ordinal classification problem of diabetic retinopathy grading classification. In order to take advantage of the ordinal essence of the problem to improve accuracy, authors propose a reasonable loss function that preserves the ordinal shape of the predicted class distribution. It is then experimentally verified to outperform existing DR methods in various settings.

**Strengths:**

- This paper focuses on a practically important problem of diabetic retinopathy grading.
- This paper recognizes the essence of the problem and accordingly propose a reasonble and feasible loss function solution.
- The proposed loss function is experimentally verified.

**Weaknesses:**

- This paper does not pay much attention to broader ordinal classification literature other than diabetic retinopathy grading.
- Questions remain for how the proposed mehtod works on other ordinal classification datasets, as well as how other ordinal classficiation methods work on DR dataset.

**Final Justification:**

I understand the focus of the paper is not only on ordinal classification but also on the diabetic retinopathy literature and is sufficiently evaluated on correctness and soundness of the paper.

**Justification:**

This is a concrete paper with a proper motivation, a well structured proposal and thorough empirical evaluation with a positive result.
The paper is overall well written and easy to follow.

---

> ### Author Rebuttal · Authors · 2025-10-22
>
> We thank the reviewer for their valuable and insightful feedback.
>
>
> **General note:**  After submission, we realized that some reported results were incorrect. We have verified and corrected them for the camera-ready version if the paper is accepted. These updates, which affect Wu-Wass, MVL, and exp_MSE, do not change the conclusions of our paper, rather strengthen our claims. The updated results (blue) are:
>
> | Experiment | AMAE ↓ | A_UOC ↓ | Kendall's τ_b ↑ | BA ↑ |
> |------------|---------:|----------:|----------------:|-----:|
> | CE         | 0.668 ± 0.060 | 0.631 ± 0.031 | 0.674 ± 0.025 | 0.518 ± 0.036 |
> | OE         | 0.653 ± 0.051 | 0.626 ± 0.029 | 0.663 ± 0.031 | 0.520 ± 0.034 |
> | BCE        | 0.659 ± 0.031 | 0.634 ± 0.018 | 0.680 ± 0.026 | 0.509 ± 0.026 |
> | UN         | 0.766 ± 0.042 | 0.675 ± 0.020 | 0.659 ± 0.033 | 0.465 ± 0.026 |
> | L_p  | *0.648 ± 0.036* | *0.622 ± 0.017* | *0.683 ± 0.017*  | *0.522 ± 0.022*  |
> | Wu-Wass    | 0.653 ± 0.035 | 0.628 ± 0.014 | 🔵 0.676 ± 0.021 | 🔵 0.520 ± 0.018 |
> | MVL        | 🔵 0.667 ± 0.034 | 🔵 0.630 ± 0.018 | 🔵 0.670 ± 0.023 | 🔵 0.518 ± 0.021 |
> | exp_MSE    | **0.616 ± 0.047** | **0.609 ± 0.027** | **0.695 ± 0.024** | 🔵 **0.535 ± 0.031** |
>
>
> - **Weakness 1:**
>   - *Reviewer's comment:* ‘This paper does not pay much attention to broader ordinal classification literature other than diabetic retinopathy grading.’
>   - **Response:** We understand this concern. While our work focuses on diabetic retinopathy, we evaluated ordinal methods originally developed for general applicability.
>
> - **Weakness 2:**
>   - *Reviewer's comment:* ‘Questions remain for how the proposed mehtod works on other ordinal classification datasets, as well as how other ordinal classficiation methods work on DR dataset.’
>   - **Response:** Although we have not tested our method on other datasets, we benchmarked it against ordinal methods developed for general use cases. For diabetic retinopathy, we evaluated the performance across datasets with differing data distributions, which is important in medical imaging.

---

### Official Review · Reviewer_UYJy · 2025-10-09
**Strong contribution to ordinal classification and medical imagery analysis**

**Rating:** 5
**Confidence:** 3

**Summary:**

This paper presents a novel ordinal loss function suited to the application of inferring Diabetic Retinopathy (DR) stage of progression, an eye condition which progresses through ordered severity levels and can be diagnosed from retinal images. The paper presents a detailed and comprehensive introduction to the task of ordinal classification and surveys existing literature proposing methods which enforce unimodality, a key criteria for well-formulated ordinal predictions, with "soft" and "hard" constraints, and contextualises the proposed approach. The paper also concisely introduces DR and the task of inferring stage of severity (from 1-5) from retinal imagery, presenting established benchmark datasets which are then used for evaluation. The experimental setup, choice of hyperparameters, choice of metrics, and results are presented clearly and succinctly and show the proposed method, called Expectation Mean Squared Error, or exp_MSE for short, performs best among considered benchmark datasets on three of four chosen metrics against considered baselines, coming in second place on the fourth metric. The paper presents a meaningful contribution to ordinal classification and validates this contribution through extensive experiments on a DR image classification task.

**Strengths:**

1. Generally clear, concise, and correct presentation: The paper presents an appropriate level of detail on DR and ordinal classification to introduce a non-expert and situate the main contributions of the work. The presentation of experimental methods and results is equally clear and a high level of detail is included in the appendix. Criteria 1 and 5 of the reviewer guidelines are thus well-satisfied.

2. Detailed summary of existing approaches to ordinal classification: A comprehensive survey of baseline approaches, and their strengths and weaknesses, is presented and properly cited. Criteria 10 and 12 are thus well-satisfied.

3. Detailed presentation and justification of evaluation metrics: Assessing ordinal regressors proves challenging as one must assess a. how correct the regressor is (accuracy on target) and also b. if the regressor respects the ordering of labels when it misses. Thus, a suite of metrics, each rewarding and penalising this dual objective slightly differently, are considered. This ties into Criteria 8 (Experimental Rigor) in the reviewer guidelines.

4. Detailed explanation of image classification task and list of benchmark datasets used for evaluation: The application considered in this submission, involving medical data, is clearly and concisely defined and the significance of achieving progress on this task is discussed, supporting evaluation Criteria 3 and 11. A small note—I do think the authors could discuss broader applications of the proposed method to other tasks more extensively—I see many potential applications in the environmental sciences, for instance.

5. Clear figures: The paper's figures efficiently communicate the intuition behind the proposed method and quickly summarise the main findings of experimental results.

6. Discussion of limitations: The authors dedicate a section openly admitting the shortcomings of the proposed method (mainly: penalising edge-cases differently than middle-cases) and brainstorming ideas (e.g. circular ordering) of how this might be addressed in future work.

**Weaknesses:**

1. Further discussion of limitations: As stated in the strengths, I commend the authors for discussing the limitation of their proposed method when dealing with edge cases. I would find it interesting to extend this discussion and to analyse, mathematically or experimentally, how the loss "falls apart" so to speak with such edge cases. Does this present a fundamental flaw when applying the method to particular problems, especially those with a small number of classes, as edge cases are proportionally large? I would imagine the loss is best-suited for ordinal classification tasks where there are a high number of classes, as edge cases are thus proportionally minimised, but this is just my intuition. Such an analysis is admittedly beyond the scope of this paper, and I do not think the authors should be penalised for not including such an analysis—rather, I mention it here as an interesting idea for future work!

2. On generalisability: The authors claim on line 206-211 that the objective is to "benchmark general-purpose ordinal methods" in this paper, and I find that to be a relatively strong claim. I do think there are generalisable takeaways from the experiments presented within and am optimistic that the proposed method would work well on other ordinal classification problems! But, I think it is bold to claim that the method "can generalise" beyond DR without any experiments beyond DR. Admittedly, this is me being a bit nitpicky on words, partially because I have so few other complaints about this paper. Perhaps the language here could be softened ever so slightly to note that the method "may generalise," or just to add a bit of nuance to acknowledge this method is only tested on one task for now. I realise this comment may seem self-contradictory, as I mentioned previously that I think the authors could discuss broader applications e.g. to environmental data, but that is not my intention.

3. Ties to CRPS could be discussed: Not really a weakness, just a comment that I wantd to make  somewhere. I am curious how the proposed method relates (or does not) to the Continuous Ranked Probability Score, commonly used to evaluate probabilistic forecasts in weather and climate. The main difference is that of course the proposed method deals with discrete data while CRPS deals with continuous data, but there feels to me a big similarity or connection to be made between these two, i.e. between CRPS and the proposed method. Perhaps the proposed method is a discretised form of CRPS? An opportunity to draw connections to other literature and methods! But, something like this is easily beyond the scope of the paper and the authors should not be penalised in any sense for leaving this sort of connection out. This is the beauty of peer review, we may use it to share ideas beyond immediate praise or critique!

**Justification:**

This paper, in my view, presents a meaningful contribution to the state of knowledge surrounding ordinal classification in machine learning, and to the state of methods applied to DR stage of severity diagnosis. The experiments are well designed, the proposed method is well introduced, contextualised, and justified, and the benchmark datasets, metrics and baselines chosen for evaluation are well documented and appropriately cited. I find the experimental results in this paper—and in the extensive appendix—to be quite convincing. Although the authors do not provide code at this time, they promise to release it upon acceptance, and otherwise present a very detailed and transparent narrative which could be used to reproduce the paper. I admit that I do not have extensive expertise in medical imagery analysis nor in ordinal classification, but I feel fairly confident in my review due to the clarity of presentation in this work.

---

> ### Author Rebuttal · Authors · 2025-10-22
>
> We thank the reviewer for their valuable and insightful feedback.
>
>
> **General note:**  After submission, we realized that some reported results were incorrect. We have verified and corrected them for the camera-ready version if the paper is accepted. These updates, which affect Wu-Wass, MVL, and exp_MSE, do not change the conclusions of our paper, rather strengthen our claims. The updated results (blue emoji) are:
>
> | Experiment | AMAE ↓ | A_UOC ↓ | Kendall's τ_b ↑ | BA ↑ |
> |------------|---------:|----------:|----------------:|-----:|
> | CE         | 0.668 ± 0.060 | 0.631 ± 0.031 | 0.674 ± 0.025 | 0.518 ± 0.036 |
> | OE         | 0.653 ± 0.051 | 0.626 ± 0.029 | 0.663 ± 0.031 | 0.520 ± 0.034 |
> | BCE        | 0.659 ± 0.031 | 0.634 ± 0.018 | 0.680 ± 0.026 | 0.509 ± 0.026 |
> | UN         | 0.766 ± 0.042 | 0.675 ± 0.020 | 0.659 ± 0.033 | 0.465 ± 0.026 |
> | L_p  | *0.648 ± 0.036* | *0.622 ± 0.017* | *0.683 ± 0.017*  | *0.522 ± 0.022*  |
> | Wu-Wass    | 0.653 ± 0.035 | 0.628 ± 0.014 | 🔵 0.676 ± 0.021 | 🔵 0.520 ± 0.018 |
> | MVL        | 🔵 0.667 ± 0.034 | 🔵 0.630 ± 0.018 | 🔵 0.670 ± 0.023 | 🔵 0.518 ± 0.021 |
> | exp_MSE    | **0.616 ± 0.047** | **0.609 ± 0.027** | **0.695 ± 0.024** | 🔵 **0.535 ± 0.031** |
>
>
> - **Weakness 1:**
>   - *Reviewer's comment:* ‘Further discussion of limitations: As stated in the strengths, I commend the authors for discussing the limitation of their proposed method when dealing with edge cases. I would find it interesting to extend this discussion and to analyse, mathematically or experimentally, how the loss "falls apart" so to speak with such edge cases. Does this present a fundamental flaw when applying the method to particular problems, especially those with a small number of classes, as edge cases are proportionally large? I would imagine the loss is best-suited for ordinal classification tasks where there are a high number of classes, as edge cases are thus proportionally minimised, but this is just my intuition. Such an analysis is admittedly beyond the scope of this paper, and I do not think the authors should be penalised for not including such an analysis—rather, I mention it here as an interesting idea for future work!’
>   - **Response:** This is indeed an interesting direction for future work, and we plan to explore it in our future research.
>
> - **Weakness 2:**
>   - *Reviewer's comment:* ‘On generalisability: The authors claim on line 206-211 that the objective is to "benchmark general-purpose ordinal methods" in this paper, and I find that to be a relatively strong claim. I do think there are generalisable takeaways from the experiments presented within and am optimistic that the proposed method would work well on other ordinal classification problems! But, I think it is bold to claim that the method "can generalise" beyond DR without any experiments beyond DR. Admittedly, this is me being a bit nitpicky on words, partially because I have so few other complaints about this paper. Perhaps the language here could be softened ever so slightly to note that the method "may generalise," or just to add a bit of nuance to acknowledge this method is only tested on one task for now. I realise this comment may seem self-contradictory, as I mentioned previously that I think the authors could discuss broader applications e.g. to environmental data, but that is not my intention.’
>   - **Response:** We understand this concern. Since our comprehensive benchmarking is limited to diabetic retinopathy, we will maintain strong claims for this application but soften the phrasing for other applications.
>
> - **Weakness 3:**
>   - *Reviewer's comment:* ‘Ties to CRPS could be discussed: Not really a weakness, just a comment that I wantd to make somewhere. I am curious how the proposed method relates (or does not) to the Continuous Ranked Probability Score, commonly used to evaluate probabilistic forecasts in weather and climate. The main difference is that of course the proposed method deals with discrete data while CRPS deals with continuous data, but there feels to me a big similarity or connection to be made between these two, i.e. between CRPS and the proposed method. Perhaps the proposed method is a discretised form of CRPS? An opportunity to draw connections to other literature and methods! But, something like this is easily beyond the scope of the paper and the authors should not be penalised in any sense for leaving this sort of connection out. This is the beauty of peer review, we may use it to share ideas beyond immediate praise or critique!’
>   - **Response:** We thank the reviewer for this suggestion. We will explore the connection between CRPS and our proposed method in future work, as we consider this direction to be beyond the scope of the current paper.

---

### Meta-Review · Area_Chair_RhVa · 2025-10-31

**Recommendation:** Accept (Poster)
**Confidence:** 4

**Metareview:**

Overall, the reviewers agree that this submission makes a valuable contribution to the characterisation of diabetic retinopathy grading. The authors have provided adequate justification and explanations of their method and its logic, as well as experiments to support it.
However, the reviewers also agree that some improvements are required and will be verified during the camera-ready submission.

These are: a) make the contributions more evident at the end of the introduction. Rather than only focusing on what this paper is doing, please position that better with respect to what this contributes to the broader body of literature. So these should read more like contributions than a summary of what is being presented in this paper; and b) enhance the paper in accordance with the responses you have provided to the reviewers.

---

### Decision · Program_Chairs · 2025-11-05

**Decision:**

Accept (Poster)

**Comment:**

We recommend a poster presentation given the AC and reviewers recommendations.